# Are Transformers with One Layer Self-Attention Using Low-Rank Weight Matrices Universal Approximators?

**Tokio Kajitsuka & Issei Sato**
Department of Computer Science
The University of Tokyo
`{kajitsuka-tokio,sato}@g.ecc.u-tokyo.ac.jp`

## Abstract

Existing analyses of the expressive capacity of Transformer models have required excessively deep layers for data memorization, leading to a discrepancy with the Transformers actually used in practice. This is primarily due to the interpretation of the softmax function as an approximation of the hardmax function. By clarifying the connection between the softmax function and the Boltzmann operator, we prove that a single layer of self-attention with low-rank weight matrices possesses the capability to perfectly capture the context of an entire input sequence. As a consequence, we show that one-layer and single-head Transformers have a memorization capacity for finite samples, and that Transformers consisting of one self-attention layer with two feed-forward neural networks are universal approximators for continuous permutation equivariant functions on a compact domain.

## 1 Introduction

The Transformer model has been ubiquitously used in deep learning since its proposal by Vaswani et al. (2017). Its widespread application spans several domains, not only revolutionizing Natural Language Processing (NLP) through models like BERT (Devlin et al., 2019; Liu et al., 2019) and GPT (Brown et al., 2020; Radford et al., a;b) but also making significant advancements in image and graph processing as an alternative to conventional models like convolutional neural networks (CNNs) and graph neural networks (GNNs) (Dosovitskiy et al., 2022; Ying et al., 2022).

One of the key reasons behind the success of the Transformer model is its ability to represent a wide range of functions. Various studies investigated this aspect, including the universal approximation theorem for Transformer models and its memorization capacity Yun et al. (2019); Kim et al. (2023); Mahdavi et al. (2023); Edelman et al. (2022); Gurevych et al. (2021); Takakura & Suzuki (2023); Likhosherstov et al. (2023).

The main challenge in proving universal approximation theorems for Transformer models lies in the fact that the Transformer needs to account for the context of the entire input sequence. Unlike feed-forward neural networks where each input is processed independently, the self-attention mechanism in Transformer models must take into account the dependencies between all elements in each input sequence. In constructive proofs (Edelman et al., 2022; Yun et al., 2019; Kim et al., 2023; Mahdavi et al., 2023; Gurevych et al., 2021; Takakura & Suzuki, 2023), these dependencies are often aggregated into a token-wise quantity, which we call a "context id" here, by a self-attention mechanism, and then feed-forward neural networks map each context id to the desired output.

The drawback of existing analyses is that they require excessively deep layers (Yun et al., 2019; Kim et al., 2023) or quite a lot of attention heads (Gurevych et al., 2021; Takakura & Suzuki, 2023; Likhosherstov et al., 2023) for data memorization, which leads to a discrepancy with Transformers being deployed in practice. This discrepancy primarily arises from the interpretation of the softmax function as an approximation of the hardmax function. Consequently, to compute the "context id" within the self-attention mechanism, the number of required self-attention parameters scales linearly with the length of an input sequence.

In this work, we address this gap by closely examining the softmax function itself. First, we show that it is impossible to output the "context id" using just one layer of self-attention with the hardmax function. At the same time, we demonstrate that just one layer of single-head and softmax-based self-attention with low-rank weight matrices possesses the capability to perfectly capture the context of an entire input sequence. This result implies that the Transformer with one self-attention layer is a universal approximator of continuous permutation equivariant functions by using two feed-forward neural networks connected before and after the single-head self-attention mechanism with an arbitrary head size.

Our contributions are summarized as follows.

1. We show that one layer self-attention with the hardmax function is not a contextual mapping; that is, one layer hardmax-based Transformer has no memorization capacity.

2. In contrast, we provide a framework for constructing a context mapping with one-layer and single-head self-attention using the softmax function.

3. We prove that one layer Transformer has a memorization capacity for finite samples, and Transformers with one-layer and single-head self-attention are universal approximators of continuous permutation equivariant functions.

## 1.1 RELATED WORKS

**Universal approximation theorems.** The history of the universal approximation theorem begins around 1990 (Cybenko, 1989; Carroll & Dickinson, 1989; Hornik, 1991; Funahashi, 1989). Recent studies on this topic include analyses of how network width and depth affect the expressive power (Lu et al., 2017), and analyses for specific architectures (Lin & Jegelka, 2018). There have also been analyses of the memorization capacity of models (Baum, 1988; Huang & Babri, 1998). The main focus of the memorization capacity is mainly on the analysis of parameter efficiency for storing finite samples (Huang, 2003; Vershynin, 2020; Park et al., 2021; Vardi et al., 2022; Yun et al., 2018; Bubeck et al., 2020; Hardt & Ma, 2016; Rajput et al., 2021; Zhang et al., 2016). Notably, Zhang et al. (2016) demonstrated that a neural network of the size used in practice can perfectly memorize a randomly labeled data set. Belkin et al. (2019); Nakkiran et al. (2019) pointed out that the minimum number of parameters required to memorize a dataset is related to the double descent threshold.

**Expressive capacity of Transformer.** Ever since Vaswani et al. (2017) first proposed the Transformer architecture, there have been various theoretical analyses on its expressive capacity. Yun et al. (2019) proved for the first time the universal approximation theorem for Transformer models, showing that a continuous function on a compact domain can be approximated if the number of Transformer blocks is on the order of the power of $n$, where $n$ is the length of each input sequence. Later, Kim et al. (2023) showed that $2n$ self-attention blocks are sufficient for the memorization of finite samples. Since the studies of Yun et al. (2019) and Kim et al. (2023) are closely related to our paper, we discuss the details in more depth in Section 3.2 later. Their results were based on the assumption that the inputs are separated to some extent, which is an assumption we also make in this paper. Alternatively, under the assumption that input sequences are linearly independent, Mahdavi et al. (2023) showed that a one-layer $H$-head self-attention mechanism can memorize $O(Hn)$ samples. Relatedly, Edelman et al. (2022) demonstrated that the bounded self-attention head is capable of expressing a sparse Boolean function while obtaining an upper bound on the covering number of self-attention. Gurevych et al. (2021) analyzed the theoretical performance of Transformers as a hierarchical composition model. Later, Takakura & Suzuki (2023) extended their result by utilizing a sinusoidal positional encoding and multiple heads, and showed that a one-layer Transformer with an embedding layer is a universal approximator for shift-equivariant $\gamma$-smooth functions. Jiang & Li (2023) recently used the Kolmogorov representation theorem to provide a non-constructive proof of the existence of a two-layer Transformer that approximates an arbitrary continuous function on a certain domain. There are variants of universal approximation theorems for Transformers, such as analyses of sparse Transformers (Yun et al., 2020) and constrained universal approximation theorems (Kratsios et al., 2021). Likhosherstov et al. (2023) showed that, given parameters, there exists an input such that self-attention approximates an arbitrary sparse pattern. While Bhojanapalli et al. (2020) proved that Transformers with a small head size, which is typical for multi-head self-attention, cannot express certain positive column-stochastic matrices, Aghajanyan et al. (2021) demonstrated empirically that pre-trained Transformers have a very low intrinsic dimension, and

Reif et al. (2019) visualized context embeddings in BERT. Luo et al. (2022) showed the existence of functions that cannot be approximated by Transformers with relative positional encoding. There is also a series of papers analyzing Transformer's expressive capabilities from the perspective of formal languages (Hahn, 2020; Bhattamishra et al., 2020; Yao et al., 2021; Hao et al., 2022; Merrill et al., 2022; Chiang & Cholak, 2022; Chiang et al., 2023), where a softmax function in a self-attention mechanism is treated as an averaging or hardmax function. As for a negative result of Transformers' expressive capacity, Zaheer et al. (2020) demonstrated that there exist some problems which sparse Transformers must require a superlinear number of layers to solve.

## 2 Preliminaries

### 2.1 Notation

We use bold lowercase letters to represent vectors and bold uppercase letters to represent matrices. For any vector $\boldsymbol{v} \in \mathbb{R}^a$, we denote by $v_i$ the $i$-the element of $\boldsymbol{v}$. For any matrix $\boldsymbol{A} \in \mathbb{R}^{a \times b}$, we denote its $i$-th row by $\boldsymbol{A}_{i,:}$, its $k$-th column by $\boldsymbol{A}_{:,k}$ and the element at its $i$-th row and $k$-th column by $A_{i,k}$. For any positive integer $m \in \mathbb{N}_+$, $[m]$ represents the set $\{1, \ldots, m\}$. For any real numbers $a < b$, $[a, b]$ represents the interval $\{x \in \mathbb{R} \mid a \leq x \leq b\}$, $(-\infty, a)$ represents $\{x \in \mathbb{R} \mid x < a\}$, and $(b, \infty)$ represents $\{x \in \mathbb{R} \mid x > b\}$. Let $\sigma_S[\boldsymbol{v}]$ and $\sigma_H[\boldsymbol{v}]$ for any input vector $\boldsymbol{v}$ be the softmax function and hardmax function, respectively. Note that when there are multiple indices with maximum values, the hardmax function is defined such that the sum of the values at these indices equals one. By abuse of notation, for any input matrix $\boldsymbol{A}$, $\sigma_S[\boldsymbol{A}]$ and $\sigma_H[\boldsymbol{A}]$ are defined as column-wise softmax and column-wise hardmax, respectively. We denote the ReLU activation function by $\sigma_R$. Unlike $\sigma_S$ and $\sigma_H$, $\sigma_R$ is always an element-wise operator, regardless of whether the input is a vector or a matrix. Let $\|\cdot\|$ be the $\ell^2$ norm and $\|\cdot\|_p$ $(1 \leq p < \infty)$ be the $\ell^p$ norm. We define the distance between two functions $f_1, f_2 : \mathbb{R}^{d \times n} \to \mathbb{R}^{d \times n}$ by

$$\mathbf{d}_p\left(f_1, f_2\right) := \left(\int \|f_1(\mathbf{X}) - f_2(\mathbf{X})\|_p^p \, \mathrm{d}\mathbf{X}\right)^{1/p}. \tag{1}$$

In this paper, $n$ denotes the length of an input sequence, $N$ the number of input sequences, $C$ the number of output classes, and $d$ the embedding dimension. In addition, $i, j$ are basically used for the indices of finite samples and $k, l$ for the indices in each input sequence.

### 2.2 Transformer block

Transformer was first introduced in Vaswani et al. (2017). Here we follow the definitions adopted in Kim et al. (2023): the Transformer block is composed of the self-attention mechanism and the feed-forward neural network, each accompanied by a skip connection. Given an input sequence $\boldsymbol{Z} \in \mathbb{R}^{d \times n}$, composed of $n$ tokens each with an embedding dimension of size $d$, a dot-product self-attention mechanism with $h$ heads outputs the following values:

$$\mathcal{F}_S^{(SA)}(\boldsymbol{Z}) = \boldsymbol{Z} + \sum_{i=1}^h \boldsymbol{W}_i^{(O)} \left(\boldsymbol{W}_i^{(V)} \boldsymbol{Z}\right) \sigma_S \left[\left(\boldsymbol{W}_i^{(K)} \boldsymbol{Z}\right)^\top \left(\boldsymbol{W}_i^{(Q)} \boldsymbol{Z}\right)\right] \in \mathbb{R}^{d \times n}, \tag{2}$$

where $\boldsymbol{W}_i^{(V)}, \boldsymbol{W}_i^{(K)}, \boldsymbol{W}_i^{(Q)} \in \mathbb{R}^{s \times d}$ and $\boldsymbol{W}_i^{(O)} \in \mathbb{R}^{d \times s}$ are the weight matrices, and $s$ is the head size. Note that here, as with Yun et al. (2019) and Kim et al. (2023), we adopt the definition of the self-attention mechanism, which excludes layer normalization from the original definition of Vaswani et al. (2017) for the sake of simplicity.

In contrast, given an input $\boldsymbol{H} \in \mathbb{R}^{d \times n}$, the output of feed-forward neural network with a skip connection at index $k \in [n]$ is

$$\mathcal{F}^{(FF)}(\boldsymbol{H})_{:,k} = \boldsymbol{H}_{:,k} + \boldsymbol{W}^{(2)} \sigma_R\left[\boldsymbol{W}^{(1)} \boldsymbol{H}_{:,k} + \boldsymbol{b}^{(1)}\right] + \boldsymbol{b}^{(2)} \in \mathbb{R}^d, \tag{3}$$

where $q$ is the hidden dimension, $\boldsymbol{W}^{(1)} \in \mathbb{R}^{q \times d}$ and $\boldsymbol{W}^{(2)} \in \mathbb{R}^{d \times q}$ are weight matrices, and $\boldsymbol{b}^{(1)} \in \mathbb{R}^q$ and $\boldsymbol{b}^{(2)}$ are bias terms.

On the basis of the above definition, the Transformer block is represented as a composition of a self-attention mechanism and a feed-forward neural network: for any input sequence $\boldsymbol{Z} \in \mathbb{R}^{d \times n}$,

composed of $n$ tokens each with an embedding dimension of size $d$, the Transformer block $\mathcal{F}$ : $\mathbb{R}^{d \times n} \to \mathbb{R}^{d \times n}$ outputs

$$\mathcal{F}(\boldsymbol{Z}) = \mathcal{F}^{(FF)}\left(\mathcal{F}_S^{(SA)}(\boldsymbol{Z})\right). \tag{4}$$

From the above definition, we see that the interaction of each token occurs only in the self-attention mechanism.

# 3 ATTENTION IS A CONTEXTUAL MAPPING

## 3.1 PROBLEM SETTING

Let $(\boldsymbol{X}^{(1)}, \boldsymbol{Y}^{(1)}), \ldots, (\boldsymbol{X}^{(1)}, \boldsymbol{Y}^{(1)}) \subset \mathbb{R}^{d \times n} \times [C]^{d \times n}$ be an $N$ input-output pairs of sequences, each of which consists of a sequence $\boldsymbol{X}^{(i)}$ of $n$ tokens with embedding dimension $d$, and an output $\boldsymbol{Y}^{(i)}$, where $\boldsymbol{Y}_{:,k}^{(i)}$ corresponds to the label of the token $\boldsymbol{X}_{:,k}^{(i)}$ at index $k$. In addition, we define the $i$-th vocabulary set for $i \in [N]$ by $\mathcal{V}^{(i)} = \bigcup_{k \in [n]} \boldsymbol{X}_{:,k}^{(i)} \subset \mathbb{R}^d$, and the whole vocabulary set $\mathcal{V}$ is defined by $\mathcal{V} = \bigcup_{i \in [N]} \mathcal{V}^{(i)} \subset \mathbb{R}^d$.

## 3.2 BACKGROUND

Yun et al. (2019) proved affirmatively one of the most fundamental questions on the expressive capacity of Transformer models, namely, whether the universal approximation theorem for Transformer models holds. Their proof approach is to quantize the input domain and reduce the universal approximation theorem to the memorization analysis of finite samples, i.e., the construction of a model that achieves zero loss for a finite number of training data, which was also analyzed later by Kim et al. (2023). In the analysis of memorization capacity, assumptions are usually made on the inputs in order to perform a meaningful analysis beyond the lower bound of Sontag (1997). Here, as with the assumptions adopted by Yun et al. (2019); Kim et al. (2023), we assume that the input tokens are separated by a certain distance.

**Definition 1** (Tokenwise Separatedness). Let $m \in \mathbb{N}$ and $\boldsymbol{Z}^{(1)}, \ldots, \boldsymbol{Z}^{(N)} \in \mathbb{R}^{m \times n}$ be input sequences. Then, $\boldsymbol{Z}^{(1)}, \ldots, \boldsymbol{Z}^{(N)}$ are called tokenwise $(r_{\min}, r_{\max}, \delta)$-separated if the following three conditions hold.

1. For any $i \in [N]$ and $k \in [n]$, $\left\|\boldsymbol{Z}_{:,k}^{(i)}\right\| > r_{\min}$ holds.

2. For any $i \in [N]$ and $k \in [n]$, $\left\|\boldsymbol{Z}_{:,k}^{(i)}\right\| < r_{\max}$ holds.

3. For any $i, j \in [N]$ and $k, l \in [n]$ with $\boldsymbol{Z}_{:,k}^{(i)} \neq \boldsymbol{Z}_{:,l}^{(j)}$, $\left\|\boldsymbol{Z}_{:,k}^{(i)} - \boldsymbol{Z}_{:,l}^{(j)}\right\| > \delta$ holds.

Note that we refer to $\boldsymbol{Z}^{(1)}, \ldots, \boldsymbol{Z}^{(N)}$ as tokenwise $(r_{\max}, \epsilon)$-separated instead if the sequences satisfy conditions 2 and 3.

The achievement of Yun et al. (2019) was not only to prove the universal approximation theorem for Transformers, but also to clarify the difficulties in the analysis of this kind of expressive capacity of Transformers and elucidated an approach to establishing the proof. Namely, what makes Transformers' memorization different from that of feed-forward neural networks is that Transformers need to capture the context of each input sequence as a whole, rather than simply associating each token with a label.

Remarkably, Yun et al. (2019); Kim et al. (2023) formulated this concept as a contextual mapping, which assigns a unique id to a pair of an input sequence and each of their tokens. We define it here using the notion of $(r, \delta)$-separatedness.

**Definition 2** (Contextual Mapping). Let $\boldsymbol{X}^{(1)}, \ldots, \boldsymbol{X}^{(N)} \in \mathbb{R}^{d \times n}$ be input sequences. Then, a map $q : \mathbb{R}^{d \times n} \to \mathbb{R}^{d \times n}$ is called an $(r, \delta)$-contextual mapping if the following two conditions hold:

1. For any $i \in [N]$ and $k \in [n]$, $\left\|q\left(\boldsymbol{X}^{(i)}\right)_{:,k}\right\| < r$ holds.

2. For any $i, j \in [N]$ and $k, l \in [n]$ such that $\mathcal{V}^{(i)} \neq \mathcal{V}^{(j)}$ or $\boldsymbol{X}_{:,k}^{(i)} \neq \boldsymbol{X}_{:,l}^{(j)}$, $\left\| q\left(\boldsymbol{X}^{(i)}\right)_{:,k} - q\left(\boldsymbol{X}^{(j)}\right)_{:,l} \right\| > \delta$ holds.

In particular, $q(\boldsymbol{X}^{(i)})$ for $i \in [N]$ is called a context id of $\boldsymbol{X}^{(i)}$.

If we have such a contextual mapping, a label sequence can be associated with a unique id for each input sequence using the existing analysis of memorization in feed-forward neural networks.

Thus, the central question is: how to construct a contextual mapping in Transformer models? The only place in Transformer models where interaction between tokens can be taken into account is in the self-attention mechanism; therefore, the self-attention mechanism must be used to construct a contextual mapping. Yun et al. (2019) first constructed a contextual mapping by using $|\mathcal{V}|^d + 1$ self-attention layers[1], and later Kim et al. (2023) improved it to $2n$ self-attention layers. However, this is still far from the practical implementation of Transformers, and it remains unclear whether a reasonably-sized Transformer would possess such memorization capacity or if the universal approximation theorem would hold. This leads to the following question.

**How many self-attention layers are both necessary and sufficient to construct a contextual mapping?**

We first point out the reason for requiring a significant number of self-attention layers in the construction of contextual mapping in the analyses of Yun et al. (2019); Kim et al. (2023). Their approach entails interpreting the softmax function in the self-attention mechanism as an approximation of the hardmax function, which also hinders a detailed analysis of the specific properties of the softmax function. As evidence of this, we illustrate in Section 3.3 that using a single layer of self-attention with the hardmax function does not suffice to construct a contextual mapping.

Next, in Section 3.4, we demonstrate that a contextual mapping can be constructed by using only one self-attention layer with the softmax function. This is somewhat surprising because this implies the probability of fully capturing the context of each input sequence only through the attention coefficients computed by the pairwise dot-product of the softmax function and its weighted average.

## 3.3 SELF-ATTENTION WITH HARDMAX

In previous studies analyzing the memorization capacity of Transformers (Yun et al., 2019; Kim et al., 2023), the softmax function is taken to be an approximation of the hardmax function. However, we show here that the attention block with the hardmax function is not a contextual mapping.

First we define the attention block with the hardmax function: for an input sequence $\boldsymbol{Z} \in \mathbb{R}^{d \times n}$, the attention with the hardmax function is calculated as

$$\mathcal{F}_H^{(SA)}(\boldsymbol{Z}) = \boldsymbol{Z} + \sum_{i=1}^{h} \boldsymbol{W}_i^{(O)} \left( \boldsymbol{W}_i^{(V)} \boldsymbol{Z} \right) \sigma_H \left[ \left( \boldsymbol{W}_i^{(K)} \boldsymbol{Z} \right)^\top \left( \boldsymbol{W}_i^{(Q)} \boldsymbol{Z} \right) \right], \qquad (5)$$

where $\boldsymbol{W}_i^{(V)}, \boldsymbol{W}_i^{(K)}, \boldsymbol{W}_i^{(Q)} \in \mathbb{R}^{s \times d}$ and $\boldsymbol{W}_i^{(O)} \in \mathbb{R}^{d \times s}$ are the weight matrices

The following theorem holds for such a model. The proof is in Appendix A.1.

**Theorem 1.** 1-*layer multi-head self-attention* $\mathcal{F}_H^{(SA)}$ *with the hardmax function cannot be a contextual mapping.*

Since the self-attention mechanism is the only place in Transformer models where interaction between tokens happens, this theorem indicates that one-layer Transformers with hardmax attention do not have a memorization capacity.

## 3.4 SELF-ATTENTION WITH SOFTMAX

In this subsection, we show that a softmax-based 1-layer attention block with low-rank weight matrices is a contextual mapping for almost all input sequences. This result is consistent with recent

---

[1]To be precise, when the continuous input range is quantized into $1/\delta$ pieces for some $0 < \delta < 1$, they demonstrated that there exists a contextual mapping composed of $\delta^{-d}$ self-attention layers.

empirical evidence that pre-trained Transformers are low-rank (Aghajanyan et al., 2021; Choromanski et al., 2020; Wang et al., 2020; Lialin et al., 2023), and theoretically supports that the low-rank self-attention mechanism is sufficient to fully comprehend the contextual information of an input sequence. It is worth noting that our construction allows for an arbitrary head size. By considering the case of a head size of 1, this particularly indicates that the self-attention mechanism has the ability to compress the information of an input sequence through a scalar value.

**Theorem 2.** *Let $\boldsymbol{X}^{(1)}, \ldots, \boldsymbol{X}^{(N)} \in \mathbb{R}^{d \times n}$ be input sequences with no duplicate word token in each sequence, that is,*

$$\boldsymbol{X}^{(i)}_{:,k} \neq \boldsymbol{X}^{(i)}_{:,l} \tag{6}$$

*for any $i \in [N]$ and $k, l \in [n]$. Also assume that $\boldsymbol{X}^{(1)}, \ldots, \boldsymbol{X}^{(N)}$ are tokenwise $(r_{\min}, r_{\max}, \epsilon)$-separated. Then, there exist weight matrices $\boldsymbol{W}^{(O)} \in \mathbb{R}^{d \times s}$ and $\boldsymbol{W}^{(V)}, \boldsymbol{W}^{(K)}, \boldsymbol{W}^{(Q)} \in \mathbb{R}^{s \times d}$ such that the ranks of $\boldsymbol{W}^{(V)}, \boldsymbol{W}^{(K)}$ and $\boldsymbol{W}^{(Q)}$ are all 1, and 1-layer single head attention with softmax, i.e., $\mathcal{F}_S^{(SA)}$ with $h = 1$ is an $(r, \delta)$-contextual mapping for the input sequences $\boldsymbol{X}^{(1)}, \ldots, \boldsymbol{X}^{(N)} \in \mathbb{R}^{d \times n}$ with $r$ and $\delta$ defined by*

$$r = r_{\max} + \frac{\epsilon}{4}, \tag{7}$$

$$\delta = \frac{2(\log n)^2 \epsilon^2 r_{\min}}{r_{\max}^2 (|\mathcal{V}| + 1)^4 (2 \log n + 3) \pi d} \exp\left(-(|\mathcal{V}| + 1)^4 \frac{(2 \log n + 3) \pi d r_{\max}^2}{4 \epsilon r_{\min}}\right). \tag{8}$$

Here we provide a simple proof sketch. The full proof can be found in Appendix A.2.

*Proof Overview.* For simplicity, we here assume $s = 1$. If we have a unique id, i.e., sequence id, corresponding to each input sequence $\boldsymbol{X}^{(i)}$ for $i \in [N]$, a context id can be constructed from a suitable linear combination of the sequence id and the value of each token. Since this linear combination can be calculated by the output projection matrix $\boldsymbol{W}^{(O)}$ and skip connection, the problem is how to configure weight parameters $\boldsymbol{W}^{(V)}, \boldsymbol{W}^{(K)}, \boldsymbol{W}^{(Q)} \in \mathbb{R}^{1 \times d}$ so that each row of the values' softmax weighted average,

$$\left(\boldsymbol{W}^{(V)} \boldsymbol{X}^{(i)}\right) \sigma_S \left[\left(\boldsymbol{W}^{(K)} \boldsymbol{X}^{(i)}\right)^\top \left(\boldsymbol{W}^{(Q)} \boldsymbol{X}^{(i)}\right)\right] \in \mathbb{R}^{1 \times n}, \tag{9}$$

outputs the unique sequence id of $\boldsymbol{X}^{(i)}$.

Actually, an even weaker condition is sufficient for an attention block to be a contextual mapping: there is no need to have just one unique sequence id for each input sequence. In fact, it is possible to construct a contextual mapping, provided that for each token $\boldsymbol{v} \in \mathcal{V}$, input sequences in which the token appears can be identified by some $\boldsymbol{v}$-dependent sequence ids. This condition can be expressed in a mathematical form as follows: what we have to show is to construct weight matrices $\boldsymbol{W}^{(V)}, \boldsymbol{W}^{(K)}, \boldsymbol{W}^{(Q)} \in \mathbb{R}^{1 \times d}$ with some $\epsilon > 0$ such that

$$\left| \left(\boldsymbol{W}^{(V)} \boldsymbol{X}^{(i)}\right) \sigma_S \left[\left(\boldsymbol{W}^{(K)} \boldsymbol{X}^{(i)}\right)^\top \left(\boldsymbol{W}^{(Q)} \boldsymbol{X}^{(i)}_{:,k}\right)\right] \right.$$
$$\left. - \left(\boldsymbol{W}^{(V)} \boldsymbol{X}^{(j)}\right) \sigma_S \left[\left(\boldsymbol{W}^{(K)} \boldsymbol{X}^{(j)}\right)^\top \left(\boldsymbol{W}^{(Q)} \boldsymbol{X}^{(j)}_{:,l}\right)\right] \right| > \epsilon \tag{10}$$

holds for any distinct $i, j \in [N]$ and any $k, l \in [n]$ such that $\boldsymbol{X}^{(i)}_{:,k} = \boldsymbol{X}^{(j)}_{:,l}$ and $\mathcal{V}^{(i)} \neq \mathcal{V}^{(j)}$.

For simplicity, we choose $\boldsymbol{W}^{(V)} = \boldsymbol{W}^{(K)} = \boldsymbol{W}^{(Q)} = \boldsymbol{w}^\top$ [2] such that the linear operator $\boldsymbol{w} \in \mathbb{R}^d$ projects each token to a scalar value while approximately preserving the distance between each pair of tokens: for any pair of tokens $\boldsymbol{v}_a, \boldsymbol{v}_b \in \mathcal{V}$,

$$c\|\boldsymbol{v}_a - \boldsymbol{v}_b\| \leq \left|\boldsymbol{w}^\top \boldsymbol{v}_a - \boldsymbol{w}^\top \boldsymbol{v}_b\right| \leq \|\boldsymbol{v}_a - \boldsymbol{v}_b\| \tag{11}$$

---

[2]In our actual proof, there exist unit vectors $\boldsymbol{v}, \boldsymbol{v}' \in \mathbb{R}^d$ such that $\boldsymbol{W}^{(V)}, \boldsymbol{W}^{(K)}$ and $\boldsymbol{W}^{(Q)}$ may be defined by $\boldsymbol{W}^{(V)} = \boldsymbol{u}''\boldsymbol{v}^\top, \boldsymbol{W}^{(K)} = \boldsymbol{u}'\boldsymbol{v}^\top$ and $\boldsymbol{W}^{(Q)} = \boldsymbol{u}\boldsymbol{v}'^\top$ for arbitrary vectors $\boldsymbol{u}, \boldsymbol{u}', \boldsymbol{u}'' \in \mathbb{R}^s$ satisfying certain constraints.

with some constant $0 < c < 1$. Then, by using the assumption $\boldsymbol{t} = \boldsymbol{X}^{(i)}_{:,k} = \boldsymbol{X}^{(j)}_{:,l}$ for some token $\boldsymbol{t} \in \mathbb{R}^d$, we have

$$
\left| \boldsymbol{w}^\top \boldsymbol{t} \right| \cdot \left| \left( \boldsymbol{w}^\top \boldsymbol{X}^{(i)} \right) \sigma_S \left[ \left( \boldsymbol{w}^\top \boldsymbol{X}^{(i)} \right)^\top \left( \boldsymbol{w}^\top \boldsymbol{t} \right) \right] - \left( \boldsymbol{w}^\top \boldsymbol{X}^{(j)} \right) \sigma_S \left[ \left( \boldsymbol{w}^\top \boldsymbol{X}^{(j)} \right)^\top \left( \boldsymbol{w}^\top \boldsymbol{t} \right) \right] \right|
$$
$$
\geq \left| \left( \boldsymbol{a}^{(i)} \right)^\top \sigma_S \left[ \boldsymbol{a}^{(i)} \right] - \left( \boldsymbol{a}^{(j)} \right)^\top \sigma_S \left[ \boldsymbol{a}^{(j)} \right] \right|, \tag{12}
$$

where we denote $\boldsymbol{a}^{(i)} = \left( \boldsymbol{w}^\top \boldsymbol{X}^{(i)} \right)^\top \left( \boldsymbol{w}^\top \boldsymbol{t} \right) \in \mathbb{R}^n$ and $\boldsymbol{a}^{(j)} = \left( \boldsymbol{w}^\top \boldsymbol{X}^{(j)} \right)^\top \left( \boldsymbol{w}^\top \boldsymbol{t} \right) \in \mathbb{R}^n$. Therefore, in order to prove that a self-attention block serves as a contextual mapping, we only have to focus on the separability of the function

$$
\mathbf{boltz} : \mathbb{R}^n \to \mathbb{R}, \boldsymbol{a} \mapsto \boldsymbol{a}^\top \sigma_S[\boldsymbol{a}], \tag{13}
$$

which is known as the Boltzmann operator (Littman, 1996; Asadi & Littman, 2017).

The following lemma shows that the Boltzmann operator is a mapping that projects input sequences to scalar values while preserving some distance, and is central to our proof that the self-attention function is a contextual mapping.

**Lemma 1.** *Let $\boldsymbol{a}^{(1)}, \ldots, \boldsymbol{a}^{(m)} \in \mathbb{R}^n$ be tokenwise $(r, \delta)$-separated vectors with no duplicate element in each vector and*

$$
\delta > 2 \log n + 3. \tag{14}
$$

*Then, the outputs of the Boltzmann operator are $(r, \delta')$-separated, that is,*

$$
\left| \mathbf{boltz}(\boldsymbol{a}^{(i)}) \right| \leq r \tag{15}
$$
$$
\left| \mathbf{boltz}(\boldsymbol{a}^{(i)}) - \mathbf{boltz}(\boldsymbol{a}^{(j)}) \right| > \delta' = (\log n)^2 e^{-2r} \tag{16}
$$

*hold for each $i, j \in [m]$ with $\boldsymbol{a}^{(i)} \neq \boldsymbol{a}^{(j)}$.*

Taking into account the above arguments, this separability of the Boltzmann operator allows us to construct one self-attention layer to be a contextual mapping. □

*Remark* 1 (Masked self-attention). In practice, attention matrices are often masked to avoid directing attention to undesired tokens. This is performed, for example, for autoregressive text generation or padding of inputs with different lengths. It is relatively straightforward to extend Theorem 2 to masked self-attention mechanisms. See Appendix C for more details.

## 4 APPLICATIONS OF CONTEXTUAL MAPPING

### 4.1 MEMORIZATION CAPACITY OF ONE-LAYER TRANSFORMER

As a first application of Theorem 2, we prove that a 1-layer Transformer can completely memorize finite samples, each of which has no duplicate token. This result emphasizes that in contrast to the proof of Kim et al. (2023), which requires $2n$ self-attention layers for Transformer memorization, one layer of self-attention is actually sufficient. In addition, it is worth noting that the hardmax-based Transformers do not have a memorization capacity, which is implied straightforwardly from Theorem 1.

**Corollary 1** (Memorization capacity of one-layer Transformer). *Let $\epsilon > 0, r_{\max} > r_{\min} > 0$ and $(\boldsymbol{X}^{(1)}, \boldsymbol{Y}^{(1)}), \ldots, (\boldsymbol{X}^{(N)}, \boldsymbol{Y}^{(N)}) \subset \mathbb{R}^{d \times n} \times [C]^{d \times n}$ be sequences of input-output-pairs such that $\boldsymbol{X}^{(1)}, \ldots, \boldsymbol{X}^{(N)}$ are tokenwise $(r_{\min}, r_{\max}, \epsilon)$-separated input sequences with no duplicate token in each sentence and consistently labeled, that is, $\boldsymbol{Y}^{(i)}_{:,k} = \boldsymbol{Y}^{(j)}_{:,l}$ holds for any $i, j \in [N]$ and $k, l \in [n]$ such that $\mathcal{V}^{(i)} = \mathcal{V}^{(j)}$ and $\boldsymbol{X}^{(i)}_{:,k} = \boldsymbol{X}^{(j)}_{:,l}$.*

*Then, there exist $4(s + d) + d(2nN + d)$ weight parameters such that for any $i \in [N]$*

$$
\mathcal{F}\left( \boldsymbol{X}^{(i)} \right) = \mathcal{F}^{(FF)}\left( \mathcal{F}^{(SA)}_S \left( \boldsymbol{X}^{(i)} \right) \right) = \boldsymbol{Y}^{(i)} \tag{17}
$$

*holds.*

*Remark* 2 (Parameter efficiency). To achieve the memorization with a one-layer Transformer, the one-hidden-layer feed-forward block has to map each context id to the corresponding label. Since the possible number of context ids is at most $nN$ in the worst case, the linear dependency on $nN$ of the number of parameters in Corollary 1 is optimal up to logarithmic factors (Bartlett et al., 2019). It is worth mentioning that this linear dependency can be relaxed to the milder requirement $\tilde{O}\left(\sqrt{nN}\right)$ by allowing for deeper layers in the feed-forward block (Vardi et al., 2022), under the assumption that the size $|\mathcal{V}|$ of the vocabulary set is independent of $n$ and $N$.

In addition, it is straightforward to show that a 1-layer Transformer with trainable positional encodings has a memorization capacity for arbitrary input sequences possibly with duplicate tokens.

**Corollary 2** (Memorization capacity of one-layer Transformer with positional encodings). *Let $\epsilon > 0, r_{\max} > r_{\min} > 0$ and $(\boldsymbol{X}^{(1)}, \boldsymbol{Y}^{(1)}), \ldots, (\boldsymbol{X}^{(N)}, \boldsymbol{Y}^{(N)}) \subset \mathbb{R}^{d \times n} \times [C]^{d \times n}$ be sequences of input-output-pairs such that $\boldsymbol{X}^{(1)}, \ldots, \boldsymbol{X}^{(N)}$ are tokenwise $(r_{\min}, r_{\max}, \epsilon)$-separated input sequences and are consistently labeled, that is, $\boldsymbol{Y}^{(i)} = \boldsymbol{Y}^{(j)}$ holds for any $i, j \in [N]$ such that $\boldsymbol{X}^{(i)} = \boldsymbol{X}^{(j)}$.*

*Then, there exist $4(s + d) + d(2nN + d)$ weight parameters and positional encodings $\boldsymbol{E} \in \mathbb{R}^{d \times n}$ such that for any $i \in [N]$,*

$$\mathcal{F}\left(\boldsymbol{X}^{(i)} + \boldsymbol{E}\right) = \mathcal{F}^{(FF)}\left(\mathcal{F}_S^{(SA)}\left(\boldsymbol{X}^{(i)} + \boldsymbol{E}\right)\right) = \boldsymbol{Y}^{(i)} \tag{18}$$

*holds.*

## 4.2 Transformers with one self-attention layer are universal approximators

As a further application of Theorem 2 we here provide a proof that Transformer with one self-attention layer is a universal approximator. More precisely, let $\mathcal{F}_{\mathrm{PE}}$ be the set of all permutation equivariant continuous functions that take values on a compact domain in $\mathbb{R}^{d \times n}$, and let $\mathcal{T}_2$ be the set of all two layer Transformers with one-layer and single-head self-attention, that is,

$$\mathcal{T}_2 = \left\{ \mathcal{F}_2^{(FF)} \circ \mathcal{F}_S^{(SA)} \circ \mathcal{F}_1^{(FF)} : \mathbb{R}^{n \times d} \to \mathbb{R}^{n \times d} \right\}, \tag{19}$$

where $\mathcal{F}_1^{(FF)}, \mathcal{F}_2^{(FF)}$ and $\mathcal{F}_S^{(SA)}$ are feed-forward neural network layers and a single-head self-attention layer with the softmax function, respectively. Then the following proposition holds (see the definition (1)).

**Proposition 1** (Transformers with one layer self-attention are universal approximators). *Let $1 \leq p < \infty$. Then, for any $f \in \mathcal{F}_{\mathrm{PE}}$ and $\epsilon > 0$, there exists a Transformer $g \in \mathcal{T}_2$ with one-layer and single-head self-attention such that $\mathbf{d}_p(f, g) < \epsilon$. holds.*

To the best of our knowledge, this is the first universal approximation theorem for two-layer Transformers with a self-attention of realistic size. Takakura & Suzuki (2023) showed that a one-layer Transformer with an embedding layer is capable of approximating shift-equivariant $\gamma$-smooth functions. However, their construction requires a high number of self-attention heads and a large head size to flatten an input sequence into outputs of self-attention. Jiang & Li (2023) used the Kolmogorov representation theorem to give a non-constructive proof of the universal approximation theorem of two-layer Transformers with positional encoding for continuous functions on a compact domain. Acknowledging the contributions of prior studies, we would like to emphasize the novelty of our results again, that is, Transformers using a single-head self-attention are universal approximators for continuous permutation equivariant functions on an arbitrary compact domain, thanks to Theorem 2. Our result can be readily extended for continuous but not necessarily permutation equivariant functions on a compact domain by using positional encoding, and at the same time is significant from the perspective of geometric deep learning.

## 5 Experiments

As shown in Theorem 2, a self-attention mechanism with rank 1 weight matrices already has enough expressive capacity to become a contextual mapping. In particular, its proof leads us to consider the following simplified form of a self-attention mechanism: for any input sequence $\boldsymbol{Z} \in \mathbb{R}^{d \times n}$,

$$\mathcal{F}_S^{(R1)}(\boldsymbol{Z}) = \boldsymbol{Z} + \boldsymbol{W}^{(O)}\left(\boldsymbol{v}_1^\top \boldsymbol{Z}\right) \sigma_S\left[\left(\boldsymbol{v}_1^\top \boldsymbol{Z}\right)^\top \left(\boldsymbol{v}_2^\top \boldsymbol{Z}\right)\right] \in \mathbb{R}^{d \times n}, \tag{20}$$

where $\boldsymbol{v}_1, \boldsymbol{v}_2 \in \mathbb{R}^d$ and $\boldsymbol{W}^{(O)} \in \mathbb{R}^{d \times 1}$ are weight matrices. This architecture corresponds to a common self-attention with the head size $s = 1$, and value and query matrices having the same weight vector $\boldsymbol{v}_1$. In this section, we test whether Transformers with self-attention layers replaced by equation 20, which we call rank-1 Transformers, actually have the theoretically predicted expressive capacity by using a real-world dataset.

We train rank-1 Transformers on a token classification task with the CoNLL-2003 (Tjong Kim Sang & De Meulder, 2003) dataset. The batch size is 32 and the training are conducted over 400 epochs.

We train three different depths of rank-1 transformers on the dataset and do not use layer normalization to match the situation with our theoretical analysis.

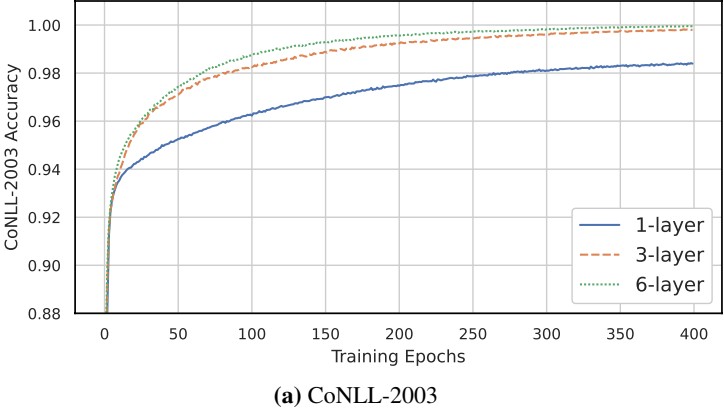

**(a)** CoNLL-2003

**Figure 1:** Training accuracy of rank-1 Transformers for the CoNLL-2003 dataset. 1-layer (solid line), 3-layer (dashed line) and 6-layer (dotted line) rank-1 Transformers are trained over 400 epochs (X-axis). For the 1-layer rank-1 Transformer, we observed that it reached an accuracy of 0.9872 at 800 epochs after further training.

Figure 1 shows training accuracies of 1-layer, 3-layer and 6-layer rank-1 Transformers on each task over 400 epochs. It can be seen that the 1-layer rank-1 Transformer is already able to memorise the CoNLL-2003 dataset almost perfectly. On the other hand, while the accuracy curve for the 1-layer rank-1 Transformer shows that the accuracy is still increasing steadily at 400 epochs, reaching 0.9872 at 800 epochs, its rate of increase is much slower than for the 3-layer and 6-layer Transformers.

From this observation, we conjecture that while theoretically 1-layer Transformers already have a memorisation capacity for finite samples, the advantage of deepening layers lies in speeding up the learning of such tasks. Since our analysis is on the expressive capabilities of Transformers, we leave this hypothesis on the optimisation aspect of Transformers as a future work.

## 6 CONCLUSIONS

We demonstrated that a contextual mapping can be implemented in one-layer and single-head self-attention with low-rank matrices, by clarifying the connection between a self-attention mechanism and the Boltzmann operator. This particularly indicates that one-layer Transformers have a memorization capacity for finite samples, and that Transformers with one-layer and single-head self-attention are universal approximators for continuous permutation equivariant functions on a compact domain. Our proof of the universal approximation theorem requires one feed-forward neural network layer before the self-attention layer to quantize continuous inputs. We leave it as future work to clarify whether the one-layer Transformers without such a quantization layer are universal approximators or not. We also expect that our analysis of the softmax function will have an impact on the evaluation of Transformer's expressive capability from the perspective of formal languages.

ACKNOWLEDGMENTS

This work was supported by JSPS KAKENHI Grant Number 20H04239 Japan. We would like to thank all the collaborators and anonymous reviewers for constructive discussions.

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

## NOTATION TABLE

**Numbers and Arrays**

$a$        A scalar

$\boldsymbol{a}$        A vector

$\boldsymbol{A}$        A matrix

$n$        The length of an input sequence

$N$        The number of input sequences

$C$        The number of output classes

$d$        Embedding dimension

$\boldsymbol{X}^{(i)}$        $i$-th input sequence, consisting of $n$ tokens of embedding dimension $d$

**Sets**

$\mathbb{R}$        Set of real numbers

$\mathbb{N}_+$        Set of positive integers

$[m]$        Set of all integers from 1 to $m$

$[a, b]$        Closed interval from $a$ to $b$

$\mathcal{V}^{(i)}$        $i$-th vocabulary set

**Indexing**

$a_i$        Element $i$ of vector $\boldsymbol{a}$, with indexing starting at 1

$A_{i,j}$        Element $i, j$ of matrix $\boldsymbol{A}$

$\boldsymbol{A}_{:,i}$        Column $i$ of matrix $\boldsymbol{A}$

$\boldsymbol{A}_{i,:}$        Row $i$ of matrix $\boldsymbol{A}$

**Functions**

$\|\boldsymbol{x}\|$ $\quad\ell^2$ norm of $\boldsymbol{x}$

$\|\boldsymbol{x}\|_p$ $\quad\ell^p$ norm of $\boldsymbol{x}$

$\mathbf{1}_{\text{condition}}$ is 1 if the condition is true, 0 otherwise

$\mathbf{d}_p(f_1, f_2)$ $\left(\int \|f_1(\mathbf{X}) - f_2(\mathbf{X})\|_p^p \, \mathrm{d}\mathbf{X}\right)^{1/p}$

$\sigma_S$ $\quad$ Softmax function

$\sigma_H$ $\quad$ Hardmax function

$\sigma_R$ $\quad$ ReLU activation function

$\mathcal{F}_H^{(SA)}$ $\quad$ Hardmax-based self-attention mechanism with a skip-connection

$\mathcal{F}_S^{(SA)}$ $\quad$ Softmax-based self-attention mechanism with a skip-connection

$\mathcal{F}^{(FF)}$ $\quad$ Feed-forward neural network with a skip-connection

**boltz** $\quad$ Boltzmann opeartor

## A PROOF OF MAIN RESULTS

First, we introduce the Boltzmann operator, which frequently appears in our proofs.

**Definition 3.** (Boltzmann operator) The Boltzmann operator is defined by

$$\mathbf{boltz} : \mathbb{R}^m \to \mathbb{R}, \ \boldsymbol{a} \mapsto \boldsymbol{a}^\top \sigma_S[\boldsymbol{a}]. \tag{21}$$

By abuse of notation, we use the same notation **boltz** for various dimension $m \in \mathbb{N}_+$.

### A.1 PROOF OF THEOREM 1

*Proof.* Let $\boldsymbol{v} \in \mathbb{R}^d$ be an arbitrary nonzero vector, and consider the situation that all input tokens can be written as

$$\mathcal{V} = \{\alpha_1 \boldsymbol{v}, \alpha_2 \boldsymbol{v}, \alpha_3 \boldsymbol{v}, \alpha_4 \boldsymbol{v}\} \subset \mathbb{R}^d \tag{22}$$

for some scalars $\alpha_1 < \alpha_2 < \alpha_3 < \alpha_4$. Then, the attention matrix inside the hardmax function at head $i$ can be expressed as

$$\left(\boldsymbol{W}_i^{(K)} \boldsymbol{v} \boldsymbol{a}^\top\right)^\top \left(\boldsymbol{W}_i^{(Q)} \boldsymbol{v} \boldsymbol{a}^\top\right) = \boldsymbol{a} \left(\boldsymbol{W}_i^{(K)} \boldsymbol{v}\right)^\top \left(\boldsymbol{W}_i^{(Q)} \boldsymbol{v}\right) \boldsymbol{a}^\top \tag{23}$$

with input coefficients $\boldsymbol{a} \in \{\alpha_1, \alpha_2, \alpha_3, \alpha_4\}^n \subset \mathbb{R}^n$. In particular, when we focus on a certain index, at which the token is, e.g., $\alpha_2 \boldsymbol{v}$, the above expression can further be written as

$$\left(\boldsymbol{W}_i^{(K)} \boldsymbol{v} \boldsymbol{a}^\top\right)^\top \left(\boldsymbol{W}_i^{(Q)} \alpha_2 \boldsymbol{v}\right) = \boldsymbol{a} \left(\boldsymbol{W}_i^{(K)} \boldsymbol{v}\right)^\top \left(\boldsymbol{W}_i^{(Q)} \boldsymbol{v}\right) \alpha_2$$
$$= \underbrace{\left(\boldsymbol{W}_i^{(K)} \boldsymbol{v}\right)^\top \left(\boldsymbol{W}_i^{(Q)} \boldsymbol{v}\right) \alpha_2}_{\in \mathbb{R}} \cdot \boldsymbol{a} \tag{24}$$

The right-hand side is a vector $\boldsymbol{a}$ multiplied by some scalar. So it is evident that the maximum value of the vector on the right-hand side is achieved only at the indices where the values of the input sequence $\boldsymbol{v}\boldsymbol{a}^\top$ are $\alpha_1$ or $\alpha_4$. This implies that a self-attention with the hardmax function invariably gets distracted by the indices where $\alpha_1$ or $\alpha_4$ are present, thereby overlooking information from other tokens in the input sequence. As a result, no matter how many heads there are, one-layer self-attention with the hardmax function cannot distinguish input sequences, e.g., $(\alpha_1 \boldsymbol{v}, \alpha_2 \boldsymbol{v}, \alpha_4 \boldsymbol{v})$ and $(\alpha_1 \boldsymbol{v}, \alpha_3 \boldsymbol{v}, \alpha_4 \boldsymbol{v})$. $\qquad\square$

### A.2 PROOF OF THEOREM 2

*Proof of Theorem 2.* Recall that a softmax-based self-attention function $\mathcal{F}_S^{(SA)} : \mathbb{R}^{d \times n} \to \mathbb{R}^{d \times n}$ with $h = 1$ is defined as

$$\mathcal{F}_S^{(SA)}(\boldsymbol{Z}) = \boldsymbol{Z} + \boldsymbol{W}^{(O)} \left(\boldsymbol{W}^{(V)} \boldsymbol{Z}\right) \sigma_S \left[\left(\boldsymbol{W}^{(K)} \boldsymbol{Z}\right)^\top \left(\boldsymbol{W}^{(Q)} \boldsymbol{Z}\right)\right], \tag{25}$$

where $\boldsymbol{W}^{(O)} \in \mathbb{R}^{d \times s}$ and $\boldsymbol{W}^{(V)}, \boldsymbol{W}^{(K)}, \boldsymbol{W}^{(Q)} \in R^{s \times d}$ are weight matrices.

We construct a softmax-based self-attention function $\mathcal{F}_S^{(SA)}$ with the property that

$$\left\| \boldsymbol{W}^{(O)} \left( \boldsymbol{W}^{(V)} \boldsymbol{X}^{(i)} \right) \sigma_S \left[ \left( \boldsymbol{W}^{(K)} \boldsymbol{X}^{(i)} \right)^\top \left( \boldsymbol{W}^{(Q)} \boldsymbol{X}_{:,k}^{(i)} \right) \right] \right\| < \frac{\epsilon}{4} \tag{26}$$

holds for any input sequence $\boldsymbol{X}^{(i)}$ with $i \in [N]$ and index $k \in [n]$. When this property is fulfilled, it is easy to show that

$$\left\| \mathcal{F}_S^{(SA)} \left( \boldsymbol{X}^{(i)} \right)_{:,k} \right\| \leq \left\| \boldsymbol{X}_{:,k}^{(i)} \right\| + \left\| \boldsymbol{W}^{(O)} \left( \boldsymbol{W}^{(V)} \boldsymbol{X}^{(i)} \right) \sigma_S \left[ \left( \boldsymbol{W}^{(K)} \boldsymbol{X}^{(i)} \right)^\top \left( \boldsymbol{W}^{(Q)} \boldsymbol{X}_{:,k}^{(i)} \right) \right] \right\|$$
$$< r_{\max} + \frac{\epsilon}{4} \tag{27}$$

holds for any $i \in [N]$ and $k \in [n]$, and also

$$\left\| \mathcal{F}_S^{(SA)} \left( \boldsymbol{X}^{(i)} \right)_{:,k} - \mathcal{F}_S^{(SA)} \left( \boldsymbol{X}^{(j)} \right)_{:,l} \right\|$$
$$\geq \left\| \boldsymbol{X}_{:,k}^{(i)} - \boldsymbol{X}_{:,l}^{(j)} \right\| - \left\| \boldsymbol{W}^{(O)} \left( \boldsymbol{W}^{(V)} \boldsymbol{X}^{(i)} \right) \sigma_S \left[ \left( \boldsymbol{W}^{(K)} \boldsymbol{X}^{(i)} \right)^\top \left( \boldsymbol{W}^{(Q)} \boldsymbol{X}_{:,k}^{(i)} \right) \right] \right\|$$
$$- \left\| \boldsymbol{W}^{(O)} \left( \boldsymbol{W}^{(V)} \boldsymbol{X}^{(j)} \right) \sigma_S \left[ \left( \boldsymbol{W}^{(K)} \boldsymbol{X}^{(j)} \right)^\top \left( \boldsymbol{W}^{(Q)} \boldsymbol{X}_{:,l}^{(j)} \right) \right] \right\|$$
$$> \epsilon - \frac{\epsilon}{4} - \frac{\epsilon}{4} = \frac{\epsilon}{2} \tag{28}$$

for any $i, j \in [N]$ and $k, l \in [n]$ such that $\boldsymbol{X}_{:,k}^{(i)} \neq \boldsymbol{X}_{:,l}^{(j)}$. So all that remains to prove is to construct a self-attention function $\mathcal{F}^{(SA)}$ that has the properties described above and can also distinguish input tokens $\boldsymbol{X}_{:,k}^{(i)} = \boldsymbol{X}_{:,l}^{(j)}$ such that $\mathcal{V}^{(i)} \neq \mathcal{V}^{(j)}$.

Let $\delta = 2 \log n + 3$ and fix any vectors $\boldsymbol{u}, \boldsymbol{u}' \in \mathbb{R}^s$ with

$$\left| \boldsymbol{u}^\top \boldsymbol{u}' \right| = (|\mathcal{V}| + 1)^4 \frac{\pi d}{8} \frac{\delta}{\epsilon r_{\min}}. \tag{29}$$

Then, according to Lemma 3 with $\delta = 2 \log n + 3$, we see that there exists a unit vector $\boldsymbol{v} \in \mathbb{R}^d$ such that

$$\left| \left( \boldsymbol{W}^{(K)} \boldsymbol{v}_a \right)^\top \left( \boldsymbol{W}^{(Q)} \boldsymbol{v}_c \right) - \left( \boldsymbol{W}^{(K)} \boldsymbol{v}_b \right)^\top \left( \boldsymbol{W}^{(Q)} \boldsymbol{v}_c \right) \right| > \delta, \tag{30}$$

$$\frac{1}{(|\mathcal{V}| + 1)^2} \sqrt{\frac{8}{\pi d}} \|\boldsymbol{v}_c\| \leq \left| \boldsymbol{v}^\top \boldsymbol{v}_c \right| \leq \|\boldsymbol{v}_c\| \tag{31}$$

for any $\boldsymbol{v}_a, \boldsymbol{v}_b, \boldsymbol{v}_c \in \mathcal{V}$ with $\boldsymbol{v}_a \neq \boldsymbol{v}_b$, where $\boldsymbol{W}^{(K)} = \boldsymbol{u} \boldsymbol{v}^\top \in \mathbb{R}^{s \times d}$ and $\boldsymbol{W}^{(Q)} = \boldsymbol{u}' \boldsymbol{v}^\top \in \mathbb{R}^{s \times d}$.

Furthermore, we configure $\boldsymbol{W}^{(O)} \in \mathbb{R}^{d \times s}$ and $\boldsymbol{W}^{(V)} \in \mathbb{R}^{s \times d}$ to be $\boldsymbol{W}^{(V)} = \boldsymbol{u}'' \boldsymbol{v}^\top$ for any nonzero vector $\boldsymbol{u}'' \in \mathbb{R}^s$ such that

$$\left\| \boldsymbol{W}^{(O)} \boldsymbol{u}'' \right\| = \frac{\epsilon}{4 r_{\max}} \tag{32}$$

holds. This can be accomplished, e.g., $\boldsymbol{W}^{(O)} = \boldsymbol{u}''' \boldsymbol{u}''^\top$ for any vector $\boldsymbol{u}''' \in \mathbb{R}^d$ which satisfies $\|\boldsymbol{u}'''\| = \epsilon / (4 r_{\max} \|\boldsymbol{u}''\|^2)$. In this case, the value of the self-attention without a skip-connection is

upper-bounded by

$$\left\| \boldsymbol{W}^{(O)} \left( \boldsymbol{W}^{(V)} \boldsymbol{X}^{(i)} \right) \sigma_S \left[ \left( \boldsymbol{W}^{(K)} \boldsymbol{X}^{(i)} \right)^\top \left( \boldsymbol{W}^{(Q)} \boldsymbol{X}^{(i)}_{:,k} \right) \right] \right\|$$

$$= \left\| \sum_{k'=1}^n s_{k'}^k \boldsymbol{W}^{(O)} \left( \boldsymbol{W}^{(V)} \boldsymbol{X}^{(i)} \right)_{:,k'} \right\| \quad \text{with } s_{k'}^k = \sigma_S \left[ \left( \boldsymbol{W}^{(K)} \boldsymbol{X}^{(i)} \right)^\top \left( \boldsymbol{W}^{(Q)} \boldsymbol{X}^{(i)}_{:,k} \right) \right]_{k'}$$

$$\leq \sum_{k'=1}^n s_{k'}^k \left\| \boldsymbol{W}^{(O)} \left( \boldsymbol{W}^{(V)} \boldsymbol{X}^{(i)} \right)_{:,k'} \right\|$$

$$\leq \max_{k' \in [n]} \left\| \boldsymbol{W}^{(O)} \left( \boldsymbol{W}^{(V)} \boldsymbol{X}^{(i)} \right)_{:,k'} \right\| \quad \text{(from } \sum_{k'=1}^n s_{k'}^k = 1)$$

$$= \max_{k' \in [n]} \left\| \boldsymbol{W}^{(O)} \boldsymbol{u}'' \boldsymbol{v}^\top \boldsymbol{X}^{(i)}_{:,k'} \right\|$$

$$= \left\| \boldsymbol{W}^{(O)} \boldsymbol{u}'' \right\| \cdot \max_{k' \in [n]} \left| \boldsymbol{v}^\top \boldsymbol{X}^{(i)}_{:,k'} \right|$$

$$\leq \frac{\epsilon}{4 r_{\max}} \cdot \max_{k' \in [n]} \left\| \boldsymbol{X}^{(i)}_{:,k'} \right\| \quad \text{(from equation 31 and equation 32)} \tag{33}$$

$$< \frac{\epsilon}{4}, \tag{34}$$

which means that equation 27 and equation 28 are satisfied with the weight matrices defined above.

Now, we see that the weight matrices $\boldsymbol{W}^{(O)}, \boldsymbol{W}^{(V)}, \boldsymbol{W}^{(K)}, \boldsymbol{W}^{(Q)}$ configured above can distinguish the most subtle pattern of input tokens, i.e. $\boldsymbol{X}^{(i)}_{:,k} = \boldsymbol{X}^{(j)}_{:,l}$ with $\mathcal{V}^{(i)} \neq \mathcal{V}^{(j)}$.

Pick up any $i, j \in [N]$ and $k, l \in [n]$ such that $\boldsymbol{X}^{(i)}_{:,k} = \boldsymbol{X}^{(j)}_{:,l}$ and $\mathcal{V}^{(i)} \neq \mathcal{V}^{(j)}$. In addition, we define $\boldsymbol{a}^{(i)}, \boldsymbol{a}^{(j)}$ by

$$\boldsymbol{a}^{(i)} = \left( \boldsymbol{W}^{(K)} \boldsymbol{X}^{(i)} \right)^\top \left( \boldsymbol{W}^{(Q)} \boldsymbol{X}^{(i)}_{:,k} \right) \in \mathbb{R}^n, \tag{35}$$

$$\boldsymbol{a}^{(j)} = \left( \boldsymbol{W}^{(K)} \boldsymbol{X}^{(j)} \right)^\top \left( \boldsymbol{W}^{(Q)} \boldsymbol{X}^{(j)}_{:,l} \right) \in \mathbb{R}^n. \tag{36}$$

Then, equation 30 and equation 31 imply that $\boldsymbol{a}^{(i)}$ and $\boldsymbol{a}^{(j)}$ are tokenwise $(r, \delta)$-separated, where $r$ is defined by

$$r = (|\mathcal{V}| + 1)^4 \frac{\pi d}{8} \frac{\delta r_{\max}^2}{\epsilon r_{\min}}, \tag{37}$$

because for any $k' \in [n]$, we have

$$\left| a^{(i)}_{k'} \right| = \left| \left( \boldsymbol{W}^{(K)} \boldsymbol{X}^{(i)}_{:,k'} \right)^\top \left( \boldsymbol{W}^{(Q)} \boldsymbol{X}^{(i)}_{:,k} \right) \right|$$

$$= \left| \left( \boldsymbol{v}^\top \boldsymbol{X}^{(i)}_{:,k'} \right)^\top \right| \cdot \left| \boldsymbol{u}^\top \boldsymbol{u}'^\top \right| \cdot \left| \left( \boldsymbol{v}^\top \boldsymbol{X}^{(i)}_{:,k} \right) \right|$$

$$\leq (|\mathcal{V}| + 1)^4 \frac{\pi d}{8} \frac{\delta}{\epsilon r_{\min}} r_{\max}^2 \quad \text{(from equation 29 and equation 31)}, \tag{38}$$

and the same upper-bound also holds for $\boldsymbol{a}^{(j)}$.

Since $\mathcal{V}^{(i)} \neq \mathcal{V}^{(j)}$ and there exists no duplicate token in $\boldsymbol{X}^{(i)}$ and $\boldsymbol{X}^{(j)}$ respectively, it follows from Lemma 1 that

$$\left| \mathbf{boltz}(\boldsymbol{a}^{(i)}) - \mathbf{boltz}(\boldsymbol{a}^{(j)}) \right| > \delta' = (\log n)^2 e^{-2r}, \tag{39}$$

that is,

$$\left| \left( \boldsymbol{a}^{(i)} \right)^\top \sigma_S \left[ \boldsymbol{a}^{(i)} \right] - \left( \boldsymbol{a}^{(j)} \right)^\top \sigma_S \left[ \boldsymbol{a}^{(j)} \right] \right| > \delta'. \tag{40}$$

Since $\boldsymbol{X}_{:,k}^{(i)} = \boldsymbol{X}_{:,l}^{(j)}$ by assumption, equation 40 are further expanded as

$$
\begin{aligned}
\delta' &< \left| \left( \boldsymbol{a}^{(i)} \right)^\top \sigma_S \left[ \boldsymbol{a}^{(i)} \right] - \left( \boldsymbol{a}^{(j)} \right)^\top \sigma_S \left[ \boldsymbol{a}^{(j)} \right] \right| \\
&= \left| \left( \boldsymbol{X}_{:,k}^{(i)} \right)^\top \left( \boldsymbol{W}^{(Q)} \right)^\top \boldsymbol{W}^{(K)} \left( \boldsymbol{X}^{(i)} \sigma_S \left[ \boldsymbol{a}^{(i)} \right] - \boldsymbol{X}^{(j)} \sigma_S \left[ \boldsymbol{a}^{(j)} \right] \right) \right| \\
&= \left| \left( \boldsymbol{X}_{:,k}^{(i)} \right)^\top \boldsymbol{v} \boldsymbol{u}'^\top \boldsymbol{u} \boldsymbol{v}^\top \left( \boldsymbol{X}^{(i)} \sigma_S \left[ \boldsymbol{a}^{(i)} \right] - \boldsymbol{X}^{(j)} \sigma_S \left[ \boldsymbol{a}^{(j)} \right] \right) \right| \\
&= \left| \boldsymbol{v}^\top \boldsymbol{X}_{:,k}^{(i)} \right| \cdot \left| \boldsymbol{u}^\top \boldsymbol{u}' \right| \cdot \left| \left( \boldsymbol{v}^\top \boldsymbol{X}^{(i)} \right) \sigma_S \left[ \boldsymbol{a}^{(i)} \right] - \left( \boldsymbol{v}^\top \boldsymbol{X}^{(j)} \right) \sigma_S \left[ \boldsymbol{a}^{(j)} \right] \right| \\
&\leq r_{\max} \cdot (|\mathcal{V}| + 1)^4 \frac{\pi d}{8} \frac{\delta}{\epsilon r_{\min}} \cdot \left| \left( \boldsymbol{v}^\top \boldsymbol{X}^{(i)} \right) \sigma_S \left[ \boldsymbol{a}^{(i)} \right] - \left( \boldsymbol{v}^\top \boldsymbol{X}^{(j)} \right) \sigma_S \left[ \boldsymbol{a}^{(j)} \right] \right|,
\end{aligned}
\tag{41}
$$

where the last inequality follows from equation 29 and equation 31.

Therefore, the gap between the outputs of th self-attention function for $\boldsymbol{X}^{(i)}$ and $\boldsymbol{X}^{(j)}$ are lower-bounded as follows:

$$
\begin{aligned}
&\left\| \mathcal{F}_S^{(SA)} \left( \boldsymbol{X}^{(i)} \right)_{:,k} - \mathcal{F}_S^{(SA)} \left( \boldsymbol{X}^{(j)} \right)_{:,l} \right\| \\
&= \left\| \boldsymbol{W}^{(O)} \left( \boldsymbol{W}^{(V)} \boldsymbol{X}^{(i)} \right) \sigma_S \left[ \boldsymbol{a}^{(i)} \right] - \boldsymbol{W}^{(O)} \left( \boldsymbol{W}^{(V)} \boldsymbol{X}^{(j)} \right) \sigma_S \left[ \boldsymbol{a}^{(j)} \right] \right\| \quad (\because \boldsymbol{X}_{:,k}^{(i)} = \boldsymbol{X}_{:,l}^{(j)}) \\
&= \left\| \boldsymbol{W}^{(O)} \boldsymbol{u}'' \right\| \cdot \left| \left( \boldsymbol{v}^\top \boldsymbol{X}^{(i)} \right) \sigma_S \left[ \boldsymbol{a}^{(i)} \right] - \left( \boldsymbol{v}^\top \boldsymbol{X}^{(j)} \right) \sigma_S \left[ \boldsymbol{a}^{(j)} \right] \right| \\
&> \frac{\epsilon}{4 r_{\max}} \cdot \frac{\delta'}{(|\mathcal{V}| + 1)^4} \frac{8 \epsilon r_{\min}}{\pi d \delta r_{\max}},
\end{aligned}
\tag{42}
$$

where $\delta$ and $\delta'$ are defined respectively as

$$
\delta = 2 \log n + 3,
\tag{43}
$$

$$
\delta' = (\log n)^2 e^{-2r} \quad \text{with} \quad r = (|\mathcal{V}| + 1)^4 \frac{\pi d}{8} \frac{\delta r_{\max}^2}{\epsilon r_{\min}}.
\tag{44}
$$

By plugging $\delta$ and $\delta'$, the above inequality is simplified as

$$
\begin{aligned}
&\left\| \mathcal{F}_S^{(SA)} \left( \boldsymbol{X}^{(i)} \right)_{:,k} - \mathcal{F}_S^{(SA)} \left( \boldsymbol{X}^{(j)} \right)_{:,l} \right\| \\
&> \frac{2 (\log n)^2 \epsilon^2 r_{\min}}{r_{\max}^2 (|\mathcal{V}| + 1)^4 (2 \log n + 3) \pi d} \exp \left( -(|\mathcal{V}| + 1)^4 \frac{(2 \log n + 3) \pi d r_{\max}^2}{4 \epsilon r_{\min}} \right).
\end{aligned}
\tag{45}
$$

$\square$

## A.3 PROOF OF COROLLARY 1

*Proof.* According to Theorem 2, we can construct such self-attention to be contextual mapping, that is, there exist weight matrices $\boldsymbol{W}^{(O)} \in \mathbb{R}^{d \times s}$ and $\boldsymbol{W}^{(V)}, \boldsymbol{W}^{(K)}, \boldsymbol{W}^{(Q)} \in \mathbb{R}^{s \times d}$ such that $\mathcal{F}_S^{(SA)}$ with $h = 1$ is a $(r, \delta)$-contextual mapping for the input sequences $\boldsymbol{X}^{(1)}, \dots, \boldsymbol{X}^{(N)}$ with $r$ and $\delta$ defined by

$$
r = r_{\max} + \frac{\epsilon}{4},
\tag{46}
$$

$$
\begin{aligned}
\delta = {}& \frac{\epsilon r_{\min} \log n}{r_{\max}^2 (|\mathcal{V}| + 1)^4 (2 \log n + 3) \pi d} \\
&\cdot \exp \left( -(|\mathcal{V}| + 1)^4 \frac{(2 \log n + 3) \pi d r_{\max}^2}{4 \epsilon r_{\min}} \right).
\end{aligned}
\tag{47}
$$

So what remains to do is to associate each context id with the corresponding output label using a feed-forward neural network $\mathcal{F}^{(FF)}$. Construction of such a network is a typical memorization

task of a one-hidden-layer feed-forward neural network. Here we adopt the implementation from Zhang et al. (2016). In this case, since the possible number of context ids is upper-bounded by $nN$, the required parameters for the FF layer with output dimension $d$ is at most $d \times (2nN + d)$ (Zhang et al., 2016). As for the self-attention layer, rank-1 weight matrices $\boldsymbol{W}^{(O)} \in \mathbb{R}^{d \times s}$ and $\boldsymbol{W}^{(V)}, \boldsymbol{W}^{(K)}, \boldsymbol{W}^{(Q)} \in \mathbb{R}^{s \times d}$ all require $s + d$ parameters each. Thus, the number of parameters of the self-attention layer is $4(s + d)$. In conclusion, the total number of parameters for one-layer Transformers to memorize the dataset is at most $4(s + d) + d(2nN + d)$. □

## A.4 PROOF OF COROLLARY 2

*Proof.* First, we define the positional encoding matrix $\boldsymbol{E} \in \mathbb{R}^{d \times n}$ as follows:

$$\boldsymbol{E} = \begin{pmatrix} 2r_{\max} & 4r_{\max} & \ldots & 2nr_{\max} \\ \vdots & \vdots & \ddots & \vdots \\ 2r_{\max} & 4r_{\max} & \ldots & 2nr_{\max} \end{pmatrix}. \tag{48}$$

Then, $\boldsymbol{X}^{(1)} + \boldsymbol{E}, \ldots, \boldsymbol{X}^{(N)} + \boldsymbol{E}$ are tokenwise $(r_{\max}, (2n+1)r_{\max}, \epsilon)$-separated, and each sentence has no duplicate token.

From Theorem 2, there exist weight matrices $\boldsymbol{W}^{(O)} \in \mathbb{R}^{d \times s}$ and $\boldsymbol{W}^V, \boldsymbol{W}^K, \boldsymbol{W}^Q \in \mathbb{R}^{s \times d}$ such that $\mathcal{F}_S^{(SA)}$ with $h = 1$ is a $(r, \delta)$-contextual mapping for the input sequences $\boldsymbol{X}^{(1)} + \boldsymbol{E}, \ldots, \boldsymbol{X}^{(N)} + \boldsymbol{E}$ with $r$ and $\delta$ defined by

$$r = (2n + 1)r_{\max} + \frac{\epsilon}{4}, \tag{49}$$

$$\delta = \frac{2(\log n)^2 \epsilon^2}{(2n + 1)^2 r_{\max}(nN + 1)^4 (2\log n + 3)\pi d}$$
$$\cdot \exp\left(-(2n + 1)^2 (nN + 1)^4 \frac{(2\log n + 3)\pi d r_{\max}}{4\epsilon}\right), \tag{50}$$

because the size of the vocabulary set of $\boldsymbol{X}^{(1)} + \boldsymbol{E}, \ldots, \boldsymbol{X}^{(N)} + \boldsymbol{E}$ is at most $nN$. Hence, hereafter we do the same thing as in the proof of Corollary 1, that is, implementing a feed-forward neural network $\mathcal{F}^{(FF)}$ which associates each context id with the corresponding label. The total number of parameters required to implement this construction can be evaluated in the same manner as in Corollary 1. □

## A.5 PROOF OF PROPOSITION 1

*Proof.* We show the propositioin by the same steps as in Yun et al. (2019). Namely,

1. First, given a permutation equivariant continuous function $f \in \mathcal{F}_{\mathrm{PE}}$ defined on a compact set, it follows from typical analysis that $f$ can be approximated by a step function with arbitrary precision in terms of $p$-norm. Therefore, to show a universal approximation theorem, it is sufficient to show that such a step function can be approximated by a Transformer with one self-attention layer.

2. Second, we use a first feed-forward neural network layer $\mathcal{F}_1^{(FF)}$ to quantize the input domain, reducing the problem to memorization of finite samples.

3. Then, by a similar analysis as in Corollary 1, it can be shown that a combination of the self-attention layer $\mathcal{F}^{(SA)}$ and $\mathcal{F}_2^{(FF)}$ can memorize the step function almost everywhere, in the sense that quantized input domains corresponding to sentences with duplicate tokens are negligibly small.

We hereafter provide rough proofs of the three steps outlined above, because there are multiple ways to construct a model that satisfies the above requirements, and we do not pursue the efficiency of feed-forward neural networks in this paper.

First, without loss of generality, we ignore skip-connections in $\mathcal{F}_1^{(FF)}$ and $\mathcal{F}_2^{(FF)}$.

1. Since $f$ is a continuous function on a compact set, $f$ has maximum and minimum values on the domain. By scaling with $\mathcal{F}_1^{(FF)}$ and $\mathcal{F}_2^{(FF)}$, $f$ is assumed to be normalized without loss of generality: for any $\boldsymbol{Z} \in \mathbb{R}^{d\times n} \setminus [0,1]^{d\times n}$

$$f(\boldsymbol{Z}) = 0, \tag{51}$$

and for any $\boldsymbol{Z} \in [-1,1]^{d \times n}$

$$-1 \leq f(\boldsymbol{Z}) \leq 1. \tag{52}$$

Let $D \in \mathbb{N}$ be the granularity of a grid

$$\mathbb{G}_D = \{1/D, 2/D, \dots, 1\}^{d\times n} \subset \mathbb{R}^{d\times n} \tag{53}$$

such that a piece-wise constant approximation

$$\overline{f}(\boldsymbol{Z}) = \sum_{\boldsymbol{L}\in\mathbb{G}_D} f(\boldsymbol{L}) \, \mathbf{1}_{\boldsymbol{Z}\in\boldsymbol{L}+[-1/D,0)^{d\times n}} \tag{54}$$

satisfies

$$\mathbf{d}_p(f, \overline{f}) < \epsilon/3. \tag{55}$$

Such a $D$ always exists because of uniform continuity of $f$.

2. We use $\mathcal{F}_1^{(FF)}$ to quantize the input domain into $\mathbb{G}_D$.

For any small $\delta > 0$, the following $\delta$-approximated step function can be constructed with one-hidden-layer feed-forward neural network: for any $z \in \mathbb{R}$

$$\frac{\sigma_R\left[z/\delta\right] - \sigma_R\left[z/\delta - 1\right]}{D} = \begin{cases} 0 & z < 0 \\ z/\delta D & 0 \leq z < \delta \\ 1/D & \delta \leq z \end{cases}. \tag{56}$$

By shifting and stacking this step function, we have an approximated multiple-step function

$$\sum_{t=0}^{D-1} \frac{\sigma_R\left[z/\delta - t/\delta D\right] - \sigma_R\left[z/\delta - 1 - t/\delta D\right]}{D}$$

$$\approx \mathbf{quant}_D(z) = \begin{cases} 0 & z < 0 \\ 1/D & 0 \leq z < 1/D \\ \vdots & \vdots \\ 1 & 1 - 1/D \leq z \end{cases}, \tag{57}$$

and subtracting the last step function from it,

$$\sum_{t=1}^{D} \frac{\sigma_R\left[z/\delta - t/\delta D\right] - \sigma_R\left[z/\delta - 1 - t/\delta D\right]}{D} - \left(\sigma_R\left[z/\delta - 1/\delta\right] - \sigma_R\left[z/\delta - 1 - 1/\delta\right]\right) \tag{58}$$

approximately quantize $[0,1]$ into $\{1/D, \dots, 1\}$, while it projects $\mathbb{R} \setminus [0,1]$ to 0.

These operations can be realized by one-hidden-layer neural network, and it is straightforward to approximate its extension $\mathbf{quant}_D$ to dimension $d \times n$, which we denote $\mathbf{quant}_D^{d\times n} : \mathbb{R}^{d\times n} \to \mathbb{R}^{d\times n}$.

In addition to that, we also add a penalty term, with which we identify whether an input sequence is in $[0,1]^{d\times n}$ or not. This is defined by

$$\sigma_R\left[(z-1)/\delta\right] - \sigma_R\left[(z-1)/\delta - 1\right] - \sigma_R\left[-z/\delta\right] - \sigma_R\left[-z/\delta - 1\right]$$

$$\approx \mathbf{penalty}(z) = \begin{cases} -1 & z \leq 0 \\ 0 & 0 < z \leq 1 \\ -1 & 1 < z \end{cases}, \tag{59}$$

which can also be implemented by one-hidden-layer feed-forward neural network.

Combining these components together, the first feed-forward neural network layer $\mathcal{F}_1^{(FF)}$ approximates the following function:

$$\overline{\mathcal{F}}_1^{(FF)}(\boldsymbol{Z}) = \mathbf{quant}_D^{d \times n}(\boldsymbol{Z}) + \sum_{t=1}^{d} \sum_{k=1}^{n} \mathbf{penalty}(\boldsymbol{Z}_{t,k}) \tag{60}$$

Note that this function quantizes inputs in $[0, 1]^{d \times n}$ with granularity $D$, while every element of the output is non-positive for inputs outside $[0, 1]^{d \times n}$. In particular, the norm of the output is upper-bounded by

$$\max_{\boldsymbol{Z} \in \mathbb{R}^{d \times n}} \left\| \mathcal{F}_1^{(FF)}(\boldsymbol{Z})_{:,k} \right\| = dn \cdot \sqrt{d} \tag{61}$$

for any $k \in [n]$.

3. Let $\tilde{\mathbb{G}}_D \subset \mathbb{G}_D$ be a sub-grid

$$\tilde{\mathbb{G}}_D = \{ \boldsymbol{L} \in \mathbb{G}_D \mid \forall k, l \in [n], \ \boldsymbol{L}_{:,k} \neq \boldsymbol{L}_{:,l} \}, \tag{62}$$

and consider memorization of $\tilde{\mathbb{G}}_D$ with its labels given by $f(\boldsymbol{L})$ for each $\boldsymbol{L} \in \tilde{\mathbb{G}}_D$. Note that the label sets are consistent because $f$ is a permutation equivariant function. Then, Theorem 2 allows us to construct a self-attention $\mathcal{F}^{(SA)}$ to be a contextual mapping for such input sequences, because $\tilde{\mathbb{G}}_D$ can be regarded as tokenwise $(1/D, \sqrt{d}, 1/D)$-separated input sequences, each of which has no duplicate token by definition. The idea is that when the granularity $D$ of $\mathbb{G}_D$ is sufficiently large, the number of cells with duplicate tokens, that is, $|\mathbb{G}_D \setminus \tilde{\mathbb{G}}_D|$ is negligible compared to the total number $|\mathbb{G}_D|$ of cells, and thus the memorization of $\tilde{\mathbb{G}}_D$ suffices for universal approximation theorem.

From the way the self-attention $\mathcal{F}^{(SA)}$ is constructed, we have

$$\left\| \mathcal{F}_S^{(SA)}(\boldsymbol{Z})_{:,k} - \boldsymbol{Z}_{:,k} \right\| < \frac{1}{4\sqrt{d}D} \max_{k' \in [n]} \|\boldsymbol{Z}_{:,k'}\| \tag{63}$$

for any $k \in [n]$ and $\boldsymbol{Z} \in \mathbb{R}^{d \times n}$. This follows from the fact that $\boldsymbol{X}^{(i)}$ in equation 33 may actually be replaced with any input sequence $\boldsymbol{Z}$, because $\boldsymbol{v}$ in equation 33 is a unit vector. In particular, combining this upper-bound with equation 61, we have

$$\left\| \mathcal{F}_S^{(SA)} \circ \mathcal{F}_1^{(FF)}(\boldsymbol{Z}_{:,k}) - \mathcal{F}^{(FF)}(\boldsymbol{Z}_{:,k}) \right\| < \frac{dn}{4D}. \tag{64}$$

Thus, if we take large enough $D$, every element of the output for $\boldsymbol{Z} \in \mathbb{R}^{d \times n} \setminus [0, 1]^{d \times n}$ is upper-bounded by

$$\mathcal{F}_S^{(SA)} \circ \mathcal{F}_1^{(FF)}(\boldsymbol{Z})_{t,k} < \frac{1}{4D} \quad (\forall t \in [d], \ k \in [n]), \tag{65}$$

while the output for $\boldsymbol{Z} \in [0, 1]^{d \times n}$ is lower-bounded by

$$\mathcal{F}_S^{(SA)} \circ \mathcal{F}_1^{(FF)}(\boldsymbol{Z})_{t,k} > \frac{3}{4D} \quad (\forall t \in [d], \ k \in [n]). \tag{66}$$

Therefore, what remains to show is construct a feed-forward neural network $\mathcal{F}_2^{(FF)}$ which associates the context id of each $\boldsymbol{L} \in \tilde{\mathbb{G}}_D \subset (3/4D, \infty)^{d \times n}$ to its corresponding label, while it outputs 0 for any input matrix $\boldsymbol{Z} \in (-\infty, 1/4D)^{d \times n}$. This can be accomplished by usual bump-functions. Precisely, construct a bump function of scale $R > 0$

$$\mathbf{bump}_R(\boldsymbol{Z}) = \frac{f(\boldsymbol{L})}{dn} \sum_{t=1}^{d} \sum_{k=1}^{n} (\sigma_R \left[ R(Z_{t,k} - L_{t,k}) - 1 \right] - \sigma_R \left[ R(Z_{t,k} - L_{t,k}) \right] \tag{67}$$

$$+ \sigma_R \left[ R(Z_{t,k} - L_{t,k}) + 1 \right]) \tag{68}$$

for each input sequence $\boldsymbol{L} \in \tilde{\mathbb{G}}_D$ and add up these functions to implement $\mathcal{F}_2^{(FF)}$.

For large enough $R > 0$, $\mathcal{F}_2^{(FF)}$ maps each input sequence $\boldsymbol{L} \in \tilde{\mathbb{G}}_D$ to its labels $f(\boldsymbol{L})$ and $\boldsymbol{Z} \in (-\infty, 1/4D)^{d \times n}$ to 0. In addition, the value of $\mathcal{F}_2^{(FF)}$ is always bounded: $0 \leq \mathcal{F}_2^{(FF)} \leq 1$. Thus, by taking sufficiently small $\delta > 0$, we have

$$\mathbf{d}_p \left( \mathcal{F}_2^{(FF)} \circ \mathcal{F}_S^{(SA)} \circ \mathcal{F}_1^{(FF)}, \mathcal{F}_2^{(FF)} \circ \mathcal{F}_S^{(SA)} \circ \overline{\mathcal{F}}_1^{(FF)} \right) < \frac{\epsilon}{3}. \tag{69}$$

Lastly, there are $|\mathbb{G}_D \setminus \tilde{\mathbb{G}}_D| = O\left(D^{-d} |\mathbb{G}_D|\right)$ input sequences with duplicate tokens, each corresponding to a cell of area $D^{-d}$. Thus, by taking sufficiently large $D$, areas of $\mathbb{G}_D \setminus \tilde{\mathbb{G}}_D$ becomes negligible and

$$\mathbf{d}_p \left( \mathcal{F}_2^{(FF)} \circ \mathcal{F}_S^{(SA)} \circ \overline{\mathcal{F}}_1^{(FF)}, \overline{f} \right) < \frac{\epsilon}{3}. \tag{70}$$

Combining equation 55, equation 69 and equation 70 together, we get the approximation error of the Transformer:

$$\mathbf{d}_p \left( \mathcal{F}_2^{(FF)} \circ \mathcal{F}_S^{(SA)} \circ \overline{\mathcal{F}}_1^{(FF)}, f \right) < \epsilon. \tag{71}$$

$\square$

# B  TECHNICAL LEMMAS

## B.1  PROJECTION OF INPUT TOKENS

We cite Lemma 13 in Park et al. (2021), with which we configure weight matrices of a self-attention mechanism.

**Lemma 2** (Park et al. (2021)). *Let $d \in \mathbb{N}$. Then, for any finite subset $\mathcal{X} \subset \mathbb{R}^d$, there exists a unit vector $\boldsymbol{v} \in \mathbb{R}^d$ such that*

$$\frac{1}{|\mathcal{X}|^2} \sqrt{\frac{8}{\pi d}} \|\boldsymbol{x} - \boldsymbol{x}'\| \leq \left| \boldsymbol{v}^\top (\boldsymbol{x} - \boldsymbol{x}') \right| \leq \|\boldsymbol{x} - \boldsymbol{x}'\| \tag{72}$$

*holds for any $\boldsymbol{x}, \boldsymbol{x}' \in \mathcal{X}$.*

The following lemma follows immediately from Lemma 2. We use $\boldsymbol{W}^{(K)}$ and $\boldsymbol{W}^{(Q)}$ in the lemma as low-rank weight matrices.[3]

**Lemma 3.** *Given $(r_{\min}, r_{\max}, \epsilon)$-separated finite vocabulary $\mathcal{V} \subset \mathbb{R}^d$ with $r_{\min} > 0$. Then, for any $\delta > 0$, there exists a unit vector $\boldsymbol{v} \in \mathbb{R}^d$ such that for any vectors $\boldsymbol{u}, \boldsymbol{u}' \in \mathbb{R}^s$ with*

$$\left| \boldsymbol{u}^\top \boldsymbol{u}' \right| = (|\mathcal{V}| + 1)^4 \frac{\pi d}{8} \frac{\delta}{\epsilon r_{\min}}, \tag{73}$$

*we have*

$$\left| \left( \boldsymbol{W}^{(K)} \boldsymbol{v}_a \right)^\top \left( \boldsymbol{W}^{(Q)} \boldsymbol{v}_c \right) - \left( \boldsymbol{W}^{(K)} \boldsymbol{v}_b \right)^\top \left( \boldsymbol{W}^{(Q)} \boldsymbol{v}_c \right) \right| > \delta, \tag{74}$$

$$\frac{1}{(|\mathcal{V}| + 1)^2} \sqrt{\frac{8}{\pi d}} \|\boldsymbol{v}_c\| \leq \left| \boldsymbol{v}^\top \boldsymbol{v}_c \right| \leq \|\boldsymbol{v}_c\| \tag{75}$$

*for any $\boldsymbol{v}_a, \boldsymbol{v}_b, \boldsymbol{v}_c \in \mathcal{V}$ with $\boldsymbol{v}_a \neq \boldsymbol{v}_b$, where $\boldsymbol{W}^{(K)} = \boldsymbol{u}\boldsymbol{v}^\top \in \mathbb{R}^{s \times d}$ and $\boldsymbol{W}^{(Q)} = \boldsymbol{u}'\boldsymbol{v}^\top \in \mathbb{R}^{s \times d}$.*

*Proof.* By applying Lemma 2 to $\mathcal{V} \cup \{0\}$, we know that there exists a unit vector $\boldsymbol{v} \in \mathbb{R}^d$ such that for any $\boldsymbol{v}_a, \boldsymbol{v}_b \in \mathcal{V} \cup \{0\}$ such that $\boldsymbol{v}_a \neq \boldsymbol{v}_b$, we have

$$\frac{1}{(|\mathcal{V}| + 1)^2} \sqrt{\frac{8}{\pi d}} \|\boldsymbol{v}_a - \boldsymbol{v}_b\| \leq \left| \boldsymbol{v}^\top (\boldsymbol{v}_a - \boldsymbol{v}_b) \right| \leq \|\boldsymbol{v}_a - \boldsymbol{v}_b\|. \tag{76}$$

---

[3]It is easy to show that different unit vectors $\boldsymbol{v}, \boldsymbol{v}' \in \mathbb{R}^d$ may be used for $\boldsymbol{W}^{(K)}$ and $\boldsymbol{W}^{(Q)}$, respectively, that is, $\boldsymbol{W}^{(K)} = \boldsymbol{u}\boldsymbol{v}^\top$ and $\boldsymbol{W}^{(Q)} = \boldsymbol{u}'\boldsymbol{v}'^\top$, as long as $\boldsymbol{v}$ and $\boldsymbol{v}'$ satisfy equation 72.

In particular, this means that for any $\boldsymbol{v}_c \in \mathcal{V}$

$$\frac{1}{(|\mathcal{V}|+1)^2} \sqrt{\frac{8}{\pi d}} \|\boldsymbol{v}_c\| \leq |\boldsymbol{v}^\top \boldsymbol{v}_c| \leq \|\boldsymbol{v}_c\| \tag{77}$$

holds. Thus, pick up arbitrary vectors $\boldsymbol{u}, \boldsymbol{u}' \in \mathbb{R}^s$ with

$$|\boldsymbol{u}^\top \boldsymbol{u}'| = (|\mathcal{V}|+1)^4 \frac{\pi d}{8} \frac{\delta}{\epsilon r_{\min}}, \tag{78}$$

and by setting $\boldsymbol{W}^{(K)} = \boldsymbol{u}\boldsymbol{v}^\top \in \mathbb{R}^{s \times d}$ and $\boldsymbol{W}^{(Q)} = \boldsymbol{u}'\boldsymbol{v}^\top \in \mathbb{R}^{s \times d}$, we have

$$\left| \left(\boldsymbol{W}^{(K)}\boldsymbol{v}_a\right)^\top \left(\boldsymbol{W}^{(Q)}\boldsymbol{v}_c\right) - \left(\boldsymbol{W}^{(K)}\boldsymbol{v}_b\right)^\top \left(\boldsymbol{W}^{(Q)}\boldsymbol{v}_c\right) \right|$$

$$= \left| (\boldsymbol{v}_a - \boldsymbol{v}_b)^\top \left(\boldsymbol{W}^{(K)}\right)^\top \left(\boldsymbol{W}^{(Q)}\boldsymbol{v}_c\right) \right|$$

$$= \left| (\boldsymbol{v}_a - \boldsymbol{v}_b)^\top \boldsymbol{v} \right| \cdot |\boldsymbol{u}^\top \boldsymbol{u}'| \cdot |\boldsymbol{v}^\top \boldsymbol{v}_c|$$

$$\geq \frac{1}{(|\mathcal{V}|+1)^2} \sqrt{\frac{8}{\pi d}} \|\boldsymbol{v}_a - \boldsymbol{v}_b\| \cdot (|\mathcal{V}|+1)^4 \frac{\pi d}{8} \frac{\delta}{\epsilon r_{\min}} \cdot \frac{1}{(|\mathcal{V}|+1)^2} \sqrt{\frac{8}{\pi d}} \|\boldsymbol{v}_c\|$$

$$> \delta, \tag{79}$$

where the last inequality follows from $(r_{\min}, r_{\max}, \epsilon)$-separatedness of $\mathcal{V}$. $\qquad \square$

## B.2 PROPERTIES OF BOLTZMANN OPERATOR

For any vector $\boldsymbol{a} = (a_1, \ldots, a_n) \in \mathbb{R}^n$, let denote $\boldsymbol{p} = (p_1, \ldots, p_n) = \sigma_S[\boldsymbol{a}]$. In addition, we define the Boltzmann operator, partition function, and entropy as follows.

$$\mathbf{boltz}(\boldsymbol{a}) = \sum_{i=1}^n a_i p_i, \tag{80}$$

$$\mathcal{L}(\boldsymbol{a}) = \log\left(\sum_{i=1}^n e^{a_i}\right), \tag{81}$$

$$\mathcal{S}(\boldsymbol{p}) = -\sum_{i=1}^n p_i \log p_i. \tag{82}$$

The following lemma shows that the Boltzmann operator is monotonically decreasing when the maximum and the rest of the arguments are far enough apart.

**Lemma 4** (Monotonicity). *Let $n \in \mathbb{N}_+$, $i \in [n]$ and $\boldsymbol{a} = (a_1, \ldots, a_n) \in \mathbb{R}^n$. Then, the derivative of the Boltzmann operator $\mathbf{boltz}(\boldsymbol{a}) = \boldsymbol{a}^\top \sigma_S[\boldsymbol{a}]$ with respect $a_i$ is*

$$\frac{\partial}{\partial a_i} \mathbf{boltz}(\boldsymbol{a}) = p_i(1 + \log p_i + \mathcal{S}(\boldsymbol{p})). \tag{83}$$

*In particular, the Boltzmann operator is monotonically decreasing with respect to $a_i$ whenever*

$$a_i < \max_{j \in [n]} a_j - \log n - 1 \tag{84}$$

*holds.*

*Proof.* Since

$$\mathcal{L}(\boldsymbol{a}) - \mathcal{S}(\boldsymbol{p}) = \log\left(\sum_{j=1}^n e^{a_j}\right) + \sum_{k=1}^n p_k \log p_k$$

$$= \sum_{k=1}^n p_k \log\left(p_k \sum_{j=1}^n e^{a_j}\right)$$

$$= \sum_{k=1}^n p_k \log e^{a_k} = \sum_{k=1}^n p_k a_k = \mathbf{boltz}(\boldsymbol{a}), \tag{85}$$

we have

$$\frac{\partial}{\partial a_i} \mathbf{boltz}(\boldsymbol{a}) = \frac{\partial}{\partial a_i} \mathcal{L}(\boldsymbol{a}) - \frac{\partial}{\partial a_i} \mathcal{S}(\boldsymbol{p}). \tag{86}$$

Notice that

$$\begin{aligned}
\frac{\partial p_i}{\partial a_j} &= \frac{\partial}{\partial a_j} \frac{e^{a_i}}{\sum_{k=1}^n e^{a_k}} \\
&= \frac{\frac{\partial}{\partial a_j} e^{a_i} \cdot \sum_{k=1}^n e^{a_k} - e^{a_i} \cdot \frac{\partial}{\partial a_j} \sum_{k=1}^n e^{a_k}}{\left(\sum_{k=1}^n e^{a_k}\right)^2} \\
&= \frac{\delta_{ij} e^{a_j} \cdot \sum_{k=1}^n e^{a_k} - e^{a_i} \cdot e^{a_j}}{\left(\sum_{k=1}^n e^{a_k}\right)^2} = p_j(\delta_{ij} - p_i) \tag{87}
\end{aligned}$$

holds for any $i, j \in [n]$. Here $\delta_{ij}$ is the Kronecker delta, that is,

$$\delta_{ij} = \begin{cases} 0 & \text{if } i \neq j, \\ 1 & \text{if } i = j. \end{cases} \tag{88}$$

Thus,

$$\frac{\partial \mathcal{L}(\boldsymbol{a})}{\partial a_i} = \frac{e^{a_i}}{\sum_{j=1}^n e^{a_j}} = p_i, \tag{89}$$

$$\begin{aligned}
\frac{\partial \mathcal{S}(\boldsymbol{p})}{\partial a_i} &= \sum_{j=1}^n \frac{\partial \mathcal{S}(\boldsymbol{p})}{\partial p_j} \cdot \frac{\partial p_j}{\partial a_i} \\
&= \sum_{j=1}^n \frac{\partial}{\partial p_j} \left(-\sum_{k=1}^n p_k \log p_k\right) \cdot p_i(\delta_{ji} - p_j) \\
&= \sum_{j=1}^n (-\log p_j - 1) \cdot p_i(\delta_{ji} - p_j) \\
&= -p_i \sum_{j=1}^n [\delta_{ji}(\log p_j + 1) - p_j \log p_j - p_j] \\
&= -p_i (\log p_i + 1 + \mathcal{S}(\boldsymbol{p}) - 1) \\
&= -p_i(\log p_i + \mathcal{S}(\boldsymbol{p})). \tag{90}
\end{aligned}$$

Plugging equation 89 and equation 90 into equation 86, we have

$$\begin{aligned}
\frac{\partial}{\partial a_i} \mathbf{boltz}(\boldsymbol{a}) &= \frac{\partial}{\partial a_i} \mathcal{L}(\boldsymbol{a}) - \frac{\partial}{\partial a_i} \mathcal{S}(\boldsymbol{p}) \\
&= p_i + p_i(\log p_i + \mathcal{S}(\boldsymbol{p})) \\
&= p_i(1 + \log p_i + \mathcal{S}(\boldsymbol{p})). \tag{91}
\end{aligned}$$

In particular, the derivative of the Boltzmann operator at index $i$ is negative when

$$\begin{aligned}
1 + \mathcal{S}(\boldsymbol{p}) + \log p_i < 0 &\Leftrightarrow 1 + \mathcal{S}(\boldsymbol{p}) + (a_i - \mathcal{L}(\boldsymbol{a})) < 0 \\
&\Leftrightarrow a_i < \mathcal{L}(\boldsymbol{a}) - \mathcal{S}(\boldsymbol{p}) - 1. \tag{92}
\end{aligned}$$

Since $\max_{j \in [n]} a_j \leq \mathcal{L}(\boldsymbol{a})$(p.72, Boyd & Vandenberghe (2004)) and $\mathcal{S}(\boldsymbol{p}) \leq \log n$, we have $\frac{\partial}{\partial a_i} \mathbf{boltz}(\boldsymbol{a}) < 0$ whenever

$$a_i < \max_{j \in [n]} a_j - \log n - 1 \tag{93}$$

holds. □

**Lemma 5** (Concavity). *Let $n \in \mathbb{N}_+$, $i \in [n]$ and $\boldsymbol{a} = (a_1, \ldots, a_n) \in \mathbb{R}^n$. Then, the Boltzmann operator $\mathbf{boltz}(\boldsymbol{a})$ is concave with respect to $a_i$, that is,*

$$\frac{\partial^2}{\partial a_i^2} \mathbf{boltz}(\boldsymbol{a}) < 0 \tag{94}$$

*holds in a domain where $\boldsymbol{a}$ satisfies*

$$a_i < \max_{j \in [n]} a_j - \log n - 3. \tag{95}$$

*Proof.* According to Lemma 4, we have

$$\frac{\partial}{\partial a_i} \mathbf{boltz}(\boldsymbol{a}) = p_i(1 + \log p_i + \mathcal{S}(\boldsymbol{p})). \tag{96}$$

Thus,

$$\begin{aligned}
\frac{\partial^2}{\partial a_i^2} \mathbf{boltz}(\boldsymbol{a}) &= \frac{\partial}{\partial a_i} \left[ p_i(1 + \log p_i + \mathcal{S}(\boldsymbol{p})) \right] \\
&= \frac{\partial p_i}{\partial a_i} \cdot (1 + \log p_i + \mathcal{S}(\boldsymbol{p})) + p_i \cdot \frac{\partial}{\partial a_i}(1 + \log p_i + \mathcal{S}(\boldsymbol{p})) \\
&= p_i(1 - p_i) \cdot (1 + \log p_i + \mathcal{S}(\boldsymbol{p})) + p_i \cdot \left[ \frac{p_i(1 - p_i)}{p_i} - p_i(\log p_i + \mathcal{S}(\boldsymbol{p})) \right] \\
&= p_i \left[ (1 - 2p_i)(\log p_i + \mathcal{S}(\boldsymbol{p}) + 1) + 1 \right],
\end{aligned} \tag{97}$$

where we used equation 87 and equation 90.

Hereafter, we show that the right-hand side of the above equality is negative under the assumption that equation 95 holds. First, the fact that $\max_{j \in [n]} a_j \leq \mathcal{L}(\boldsymbol{a})$ (p.72, Boyd & Vandenberghe (2004)) and $\mathcal{S}(\boldsymbol{p}) \leq \log n$ implies

$$a_i < \max_{j \in [n]} a_j - \log n - 3 < \mathcal{L}(\boldsymbol{a}) - \mathcal{S}(\boldsymbol{p}) - 3. \tag{98}$$

It follows from $a_i - \mathcal{L}(\boldsymbol{a}) = \log p_i$ that

$$a_i < \max_{j \in [n]} a_j - \log n - 3 \Rightarrow \log p_i < -\mathcal{S}(\boldsymbol{p}) - 3. \tag{99}$$

Next, since the entropy $\mathcal{S}(\boldsymbol{p})$ is always non-negative, we have $\log p_i < -3$ under equation 95. By using an inequality $e^{-x} \leq \frac{1}{1+x}$ for $x > -1$, this implies

$$p_i < e^{-3} \leq \frac{1}{1 + 3} = \frac{1}{4}. \tag{100}$$

Thus, as long as equation 95 holds, we have

$$(1 - 2p_i)(\log p_i + \mathcal{S}(\boldsymbol{p}) + 1) + 1 < \left( 1 - 2 \cdot \frac{1}{4} \right) \cdot (-2) + 1 = 0, \tag{101}$$

which in turn implies $\frac{\partial^2}{\partial a_i^2} \mathbf{boltz}(\boldsymbol{a}) < 0$. $\square$

**Lemma 6.** *Let $n \geq 2$ and $\boldsymbol{a} = (a_1, \ldots, a_{n-1}, a_n), \boldsymbol{b} = (b_1, \ldots b_{n-1}, b_n) \in \mathbb{R}^n$ be sequences such that the first $n - 1$ elements of $\boldsymbol{a}$ and $\boldsymbol{b}$ match, that is, $a_i = b_i$ for all $i \in [n-1]$. In addition, if*

$$\max_{i \in [n-1]} a_i - \delta > a_n > b_n \tag{102}$$

*with $\delta > \log n + 3$, the difference $\mathbf{boltz}(\boldsymbol{a}) - \mathbf{boltz}(\boldsymbol{b})$ is lower-bounded by*

$$\mathbf{boltz}(\boldsymbol{b}) - \mathbf{boltz}(\boldsymbol{a}) > (a_n - b_n)(\delta + a_n - b_n - \log n - 1) \cdot \frac{e^{b_n}}{\sum_{i=1}^{n} e^{b_i}}. \tag{103}$$

*Proof.* According to the monotonicity (Lemma 4) and concavity (Lemma 5) of the Boltzmann operator, we have

$$\mathbf{boltz}(b_1, \ldots, b_{n-1}, x) + (a_n - x) \cdot \frac{\partial}{\partial x} \mathbf{boltz}(b_1, \ldots, b_{n-1}, x) > \mathbf{boltz}(b_1, \ldots, b_{n-1}, a_n) \tag{104}$$

for any $x < a_n$. In particular, by setting $x = b_n$ and using equation 83, this implies

$$\begin{aligned}
\mathbf{boltz}(\boldsymbol{b}) - \mathbf{boltz}(\boldsymbol{a}) &> (a_n - b_n) \cdot \left( - \frac{\partial}{\partial x} \mathbf{boltz}(b_1, \ldots, b_{n-1}, x) \Big|_{x=b_n} \right) \\
&= (a_n - b_n) \cdot \left[ -p_n(1 + \log p_n + \mathcal{S}(\boldsymbol{p})) \right] \\
&> (a_n - b_n) \cdot \left[ -p_n \left( 1 + b_n - \max_{i \in [n]} b_i + \log n \right) \right] \\
&> (a_n - b_n) \cdot p_n(\delta + a_n - b_n - \log n - 1),
\end{aligned} \tag{105}$$

where $\boldsymbol{p} = (p_1, \ldots, p_n) \in \mathbb{R}^n$ is the softmax function of $\boldsymbol{b}$, i.e., $\boldsymbol{p} = \sigma_S[\boldsymbol{b}]$. $\square$

### B.3 Proof of Lemma 1

Before moving on to the proof, we first illustrate the proof sketch of Lemma 1 using a simple example.

Since the Boltzmann operator is permutation invariant, without loss of generality we assume the elements of $\boldsymbol{a}^{(i)}$ and $\boldsymbol{a}^{(j)}$ are sorted in descending order, e.g., $\boldsymbol{a}^{(i)} = (8, 6, 5)$ and $\boldsymbol{a}^{(j)} = (4, 3, 1)$. In this case, since the Boltzmann operator **boltz** can be regarded as an approximation of $\max$, we have

$$\mathbf{boltz}(\boldsymbol{a}^{(i)}) \approx 8 > 4 \approx \mathbf{boltz}(\boldsymbol{a}^{(j)}), \tag{106}$$

and so the Boltzmann operator can readily separate these two inputs.

The subtle case is where the initial tokens of $a^{(i)}$ and $a^{(j)}$ are identical, but the rest of each sequences differs, like $\boldsymbol{a}^{(i)} = (8, 6, 5)$ and $\boldsymbol{a}^{(j)} = (8, 3, 1)$. However, a closer observation reveals that if the first coordinate and the second one of the input $\boldsymbol{a} \in \mathbb{R}^n$ are well-separated, then $\mathbf{boltz}(\boldsymbol{a})$ is monotonically decreasing for each coordinate $k = 2, \ldots, n$. In the above example, this intuitively implies

$$\mathbf{boltz}\left(\boldsymbol{a}^{(j)}\right) \geq \mathbf{boltz}\left(\overline{\boldsymbol{a}}^{(j)}\right) \quad \text{with } \overline{\boldsymbol{a}}^{(j)} = (8, 3, 3) \tag{107}$$

and

$$\mathbf{boltz}\left(\boldsymbol{a}^{(i)}\right) \leq \mathbf{boltz}\left(\underline{\boldsymbol{a}}^{(i)}\right) \quad \text{with } \underline{\boldsymbol{a}}^{(i)} = (8, 6, -\infty), \tag{108}$$

or by abuse of notation, $\mathbf{boltz}\left(\boldsymbol{a}^{(i)}\right) \leq \mathbf{boltz}\left(8, 6\right)$.

Thus, if we can show $\mathbf{boltz}\left(\underline{\boldsymbol{a}}^{(i)}\right) < \mathbf{boltz}\left(\overline{\boldsymbol{a}}^{(j)}\right)$, then $\mathbf{boltz}\left(\boldsymbol{a}^{(i)}\right) < \mathbf{boltz}\left(\boldsymbol{a}^{(j)}\right)$ holds and we know that the Boltzmann operator can also separate this pattern of inputs. Fortunately, this is indeed the case if each element of the inputs is sufficiently separated, because

$$\mathbf{boltz}\left(\overline{\boldsymbol{a}}^{(j)}\right) = \frac{8e^8 + 3e^3 + 3e^3}{e^8 + e^3 + e^3} = \frac{8e^8 + 2 \cdot 3e^3}{e^8 + 2 \cdot e^3}$$
$$\approx \frac{8e^8 + (3 + \log 2)e^{3+\log 2}}{e^8 + \cdot e^{3+\log 2}} = \mathbf{boltz}\left(8, 3 + \log 2\right), \tag{109}$$

and we have $\mathbf{boltz}\left(8, 6\right) < \mathbf{boltz}\left(8, 3 + \log 2\right)$ by the monotonicity of the Boltzmann operator.

*Proof of Lemma 1.* We only have to show the case of $m = 2$, and for notational convenience, $\boldsymbol{a}^{(1)}$ (resp. $\boldsymbol{a}^{(2)}$) hereafter is denoted by $\boldsymbol{a}$ (resp. $\boldsymbol{b}$). Also, since the Boltzmann operator is permutation invariant, we assume without loss of generality $a_1 > \cdots > a_n$ and $b_1 > \cdots > b_n$ (Since there is no duplicate element in $\boldsymbol{a}$, $a_k$ is strictly greater than $a_l$ for any $k < l$. The same holds for $\boldsymbol{b}$).

First, since the Boltzmann operator can be regarded as weighting averaging, we have

$$|\mathbf{boltz}(\boldsymbol{a})| \leq \max(|a_1|, |a_n|) < r. \tag{110}$$

For $\delta'$-separatedness, if $\boldsymbol{a} \neq \boldsymbol{b}$, w.l.o.g. we assume that there exists $k \in \{0, \ldots, n-1\}$ such that

$$(a_1, \ldots, a_k) = (b_1, \ldots, b_k) \text{ and } a_{k+1} > b_{k+1}. \tag{111}$$

Then, Lemma 7 implies that we have

$$|\mathbf{boltz}(\boldsymbol{a}) - \mathbf{boltz}(\boldsymbol{b})| > (\log n)^2 e^{-(a_1 - b_{k+1})}. \tag{112}$$

Since $\boldsymbol{a}$ and $\boldsymbol{b}$ are tokenwise $(r, \delta)$-separated, $a_1 - b_{k+1} < 2r$ holds. Thus, the right-hand side of the above inequality is further lower-bounded by

$$|\mathbf{boltz}(\boldsymbol{a}) - \mathbf{boltz}(\boldsymbol{b})| > (\log n)^2 e^{-(a_1 - b_{k+1})} > (\log n)^2 e^{-2r}. \tag{113}$$

$\square$

**Lemma 7.** *Let $\boldsymbol{a}, \boldsymbol{b} \in \mathbb{R}^n$ be tokenwise $\delta$-separated vectors in a decreasing order with no duplicate element in each vector and $\delta > 2\log n + 3$, that is,*

$$i > j \Rightarrow a_i - a_j, b_i - b_j > \delta, \tag{114}$$
$$a_i \neq b_j \Rightarrow |a_i - b_j| > \delta \tag{115}$$

*for any $i, j \in [n]$.*

*In addition, suppose there exists $k \in \{0, \ldots, n-1\}$ such that*

$$(a_1, \ldots, a_k) = (b_1, \ldots, b_k) \text{ and } a_{k+1} > b_{k+1}. \tag{116}$$

*Then, the outputs of the Boltzmann operator are $(\log n)^2 e^{-(a_1 - b_{k+1})}$-separated, that is,*

$$|\mathbf{boltz}(\boldsymbol{a}) - \mathbf{boltz}(\boldsymbol{b})| > (\log n)^2 e^{-(a_1 - b_{k+1})} \tag{117}$$

*holds.*

*Proof.* We show the lemma by dividing it into the following two cases:

1. $k \geq 1$

   Let $\underline{\boldsymbol{a}}$ and $\overline{\boldsymbol{b}}$ be

   $$\underline{\boldsymbol{a}} = (a_1, a_2, \ldots, a_k, a_{k+1}) \in \mathbb{R}^{k+1} \tag{118}$$
   $$\overline{\boldsymbol{b}} = (a_1, a_2, \cdots, a_k, b_{k+1}, b_{k+1}, \ldots, b_{k+1}) \in \mathbb{R}^n. \tag{119}$$

   Then, by abuse of notation, Lemma 4 implies that

   $$\mathbf{boltz}(\boldsymbol{a}) < \underline{\boldsymbol{a}}^\top \sigma_S[\underline{\boldsymbol{a}}] = \mathbf{boltz}(\underline{\boldsymbol{a}}) \quad \text{and} \quad \mathbf{boltz}(\boldsymbol{b}) > \mathbf{boltz}(\overline{\boldsymbol{b}}), \tag{120}$$

   and we only have to evaluate the magnitude of the difference $\mathbf{boltz}(\overline{\boldsymbol{b}}) - \mathbf{boltz}(\underline{\boldsymbol{a}})$.

   Let $\gamma_k$ and $\xi_k$ be

   $$\gamma_k = \sum_{l=1}^{k} a_l e^{a_l} \quad \text{and} \quad \xi_k = \sum_{l=1}^{k} e^{a_l}. \tag{121}$$

   Then, $\mathbf{boltz}(\overline{\boldsymbol{b}})$ can be decomposed into

   $$\begin{aligned}
   \mathbf{boltz}(\overline{\boldsymbol{b}}) &= \frac{\gamma_k + (n-k)b_{k+1}e^{b_{k+1}}}{\xi_k + (n-k)e^{b_{k+1}}} \\
   &= \frac{\gamma_k + b_{k+1}e^{b_{k+1}+\log(n-k)}}{\xi_k + e^{b_{k+1}+\log(n-k)}} \\
   &= \frac{\gamma_k + (b_{k+1} + \log(n-k))e^{b_{k+1}+\log(n-k)}}{\xi_k + e^{b_{k+1}+\log(n-k)}} - \frac{\log(n-k) \cdot e^{b_{k+1}+\log(n-k)}}{\xi_k + e^{b_{k+1}+\log(n-k)}} \\
   &= \mathbf{boltz}(a_1, \ldots, a_k, b_{k+1} + \log(n-k)) - \frac{\log(n-k) \cdot e^{b_{k+1}+\log(n-k)}}{\xi_k + e^{b_{k+1}+\log(n-k)}}.
   \end{aligned} \tag{122}$$

   Therefore, the difference $\mathbf{boltz}(\overline{\boldsymbol{b}}) - \mathbf{boltz}(\underline{\boldsymbol{a}})$ can be written as

   $$\begin{aligned}
   \mathbf{boltz}(\overline{\boldsymbol{b}}) - \mathbf{boltz}(\underline{\boldsymbol{a}}) &= \mathbf{boltz}(a_1, \ldots, a_k, b_{k+1} + \log(n-k)) - \mathbf{boltz}(\underline{\boldsymbol{a}}) \\
   &\quad - \frac{\log(n-k) \cdot e^{b_{k+1}+\log(n-k)}}{\xi_k + e^{b_{k+1}+\log(n-k)}}.
   \end{aligned} \tag{123}$$

   According to Lemma 6, the first two terms on the right-hand side are evaluated as

   $$\begin{aligned}
   &\mathbf{boltz}(a_1, \ldots, a_k, b_{k+1} + \log(n-k)) - \mathbf{boltz}(\underline{\boldsymbol{a}}) \\
   &> (a_{k+1} - b_{k+1} - \log(n-k))(\delta + a_{k+1} - b_{k+1} - \log(n-k) - \log(k+1) - 1) \\
   &\quad \cdot \frac{e^{b_{k+1}+\log(n-k)}}{\xi_k + e^{b_{k+1}+\log(n-k)}} \\
   &> (\delta - \log n)(2\delta - 2\log n - 1) \cdot \frac{e^{b_{k+1}+\log(n-k)}}{\xi_k + e^{b_{k+1}+\log(n-k)}}.
   \end{aligned} \tag{124}$$

Since $\delta > 2\log n + 3$ by assumption, the above inequality is further lower-bounded by

$$\mathbf{boltz}\,(a_1,\ldots,a_k,b_{k+1}+\log(n-k)) - \mathbf{boltz}\,(\underline{a})$$
$$> (\delta - \log n)(2\delta - 2\log n - 1) \cdot \frac{e^{b_{k+1}+\log(n-k)}}{\xi_k + e^{b_{k+1}+\log(n-k)}}$$
$$> (\log n + 3)(2\log n + 5) \cdot \frac{e^{b_{k+1}+\log(n-k)}}{\xi_k + e^{b_{k+1}+\log(n-k)}}. \tag{125}$$

Plugging the above inequality into equation 123, we see

$$\mathbf{boltz}\,(\overline{b}) - \mathbf{boltz}\,(\underline{a}) = \mathbf{boltz}\,(a_1,\ldots,a_k,b_{k+1}+\log(n-k)) - \mathbf{boltz}\,(\underline{a})$$
$$- \frac{\log(n-k) \cdot e^{b_{k+1}+\log(n-k)}}{\xi_k + e^{b_{k+1}+\log(n-k)}}$$
$$> \frac{e^{b_{k+1}+\log(n-k)}}{\xi_k + e^{b_{k+1}+\log(n-k)}}\left[(\log n + 3)(2\log n + 5) - \log(n-k)\right]$$
$$> \frac{e^{b_{k+1}+\log(n-k)}}{\xi_k + e^{b_{k+1}+\log(n-k)}} \cdot 2(\log n)^2. \tag{126}$$

Lastly, notice that the following inequality follows from $\delta$-separatedness of $a$ and $b$ and the assumption that $a$ has no duplicate token:

$$\xi_k + e^{b_{k+1}+\log(n-k)} < \sum_{l=1}^{k+1} e^{a_l} \quad (\because a_{k+1} > b_{k+1} + \log(n-k))$$
$$< e^{a_1} \sum_{l=1}^{k+1} e^{-(l-1)\delta}$$
$$< 2e^{a_1} \quad (\because \delta > \log 2). \tag{127}$$

By using this inequality, equation 126 is lower-bounded by

$$\mathbf{boltz}\,(\overline{b}) - \mathbf{boltz}\,(\underline{a}) > \frac{e^{b_{k+1}+\log(n-k)}}{\xi_k + e^{b_{k+1}+\log(n-k)}} \cdot 2(\log n)^2$$
$$> \frac{e^{b_{k+1}+\log(n-k)}}{2e^{a_1}} \cdot 2(\log n)^2$$
$$> (\log n)^2 e^{-(a_1 - b_{k+1})}, \tag{128}$$

which implies $\mathbf{boltz}(b) - \mathbf{boltz}(a) > (\log n)^2 e^{-(a_1 - b_{k+1})}$.

2. $k = 0$

   Since the Boltzmann operator can be regarded as weighted averaging, $\mathbf{boltz}(b) \leq b_1$ always holds. Thus, it is enough to evaluate how much greater $\mathbf{boltz}(a)$ is than $b_1$.

   Let $\overline{a} \in \mathbb{R}^n$ be

   $$\overline{a} = (a_1, a_1 - \delta, \ldots, a_1 - \delta). \tag{129}$$

   Then, $\mathbf{boltz}(a) > \mathbf{boltz}(\overline{a})$ follows from Lemma 4, and its value is

   $$\mathbf{boltz}\,(\overline{a}) = \frac{a_1 e^{a_1} + (n-1)(a_1 - \delta)e^{a_1-\delta}}{e^{a_1} + (n-1)e^{a_1-\delta}}$$
   $$= \frac{a_1 + (n-1)(a_1 - \delta)e^{-\delta}}{1 + (n-1)e^{-\delta}}$$
   $$= a_1 - \frac{(n-1)\delta e^{-\delta}}{1 + (n-1)e^{-\delta}}. \tag{130}$$

Therefore, the difference $\mathbf{boltz}\,(\boldsymbol{a}) - \mathbf{boltz}\,(\boldsymbol{b})$ is

$$
\begin{aligned}
\mathbf{boltz}\,(\boldsymbol{a}) - \mathbf{boltz}\,(\boldsymbol{b}) &\geq \mathbf{boltz}\,(\overline{\boldsymbol{a}}) - b_1 \\
&> a_1 - \frac{(n-1)\delta e^{-\delta}}{1 + (n-1)e^{-\delta}} - (a_1 - \delta) \\
&= \delta - \frac{(n-1)\delta e^{-\delta}}{1 + (n-1)e^{-\delta}} \\
&= \frac{\delta}{1 + (n-1)e^{-\delta}} \\
&\geq \log n,
\end{aligned}
\tag{131}
$$

where the last inequality follows from the assumption $\delta > 2\log n$. Note that the right-hand side is greater than $(\log n)^2 e^{-(a_1 - b_1)}$, because $a_1 - b_1 > \log n$ implies $\log n \cdot e^{-(a_1 - b_1)} < 1$.

$\square$

## C  EXTENSION TO MASKED SELF-ATTENTION

Masked self-attention mechanisms are formulated as follows: the Softmax part in equation 2 is replaced by

$$
\sigma_S\left[\left(\boldsymbol{W}_{l,i}^{(K)}\boldsymbol{Z}\right)^{\top}\left(\boldsymbol{W}_{l,i}^{(Q)}\boldsymbol{Z}\right) + \boldsymbol{C}\right] \in \mathbb{R}^{d \times n},
\tag{132}
$$

with some masking matrix $\boldsymbol{C} \in \mathbb{R}^{n \times n}$ whose elements are either $0$ or $-\infty$.

Our main result Theorem 2 can be readily extended to the case where attention masking are conducted. The idea is that

$$
\begin{aligned}
\mathbf{boltz}(\boldsymbol{a} + \boldsymbol{c}) &= (\boldsymbol{a} + \boldsymbol{c})^{\top}\sigma_S\,[\boldsymbol{a} + \boldsymbol{c}] \\
&= \boldsymbol{a}^{\top}\sigma_S\,[\boldsymbol{a} + \boldsymbol{c}] + \boldsymbol{c}^{\top}\sigma_S\,[\boldsymbol{a} + \boldsymbol{c}] \\
&= \boldsymbol{a}^{\top}\sigma_S\,[\boldsymbol{a} + \boldsymbol{c}]
\end{aligned}
\tag{133}
$$

holds for any masking vector $\boldsymbol{c} \in \mathbb{R}^n$ whose elements are either $0$ or $-\infty$. Thus, in order to ensure that the masked attention is a contextual mapping, it is sufficient to verify $\mathbf{boltz}(\boldsymbol{a} + \boldsymbol{c})$ are well-separated. The caveat is that the Boltzmann operator now has to distinguish inputs consisting of the same attention $\boldsymbol{a}$, but different masks $\boldsymbol{c}_1, \boldsymbol{c}_2$. However, the separability in this case can also be proved in the same way as in the proof of Lemma 1.

