# OpenReview forum: "Are Transformers with One Layer Self-Attention Using Low-Rank Weight Matrices Universal Approximators?"
_ICLR.cc/2024/Conference — ICLR 2024 poster_

### Official Review · Reviewer_mvi8 · 2023-10-31

**Soundness:** 3 good
**Presentation:** 3 good
**Contribution:** 2 fair
**Rating:** 6
**Confidence:** 4

**Summary:**

This paper proves that Transformers with one self-attention layer using low-rank weight matrices, plus two FFNs, are universal appoximators of sequence-to-sequence functions. It makes a connection between the softmax function and the Boltzmann operator, and argues that softmax attention can be more useful for universal approximation than hardmax which is commonly used in existing papers.

**Strengths:**

- I appreciate the connection between the softmax function and the Boltzmann operator.

- The negative result on hardmax attention looks interesting to me.

**Weaknesses:**

- **Significance of the result seems limited to me.** Some claims in the paper are weak or wrong.
   - The authors claim that existing results on the expressiveness of Transformers require excessively deep layers or a lot of attention heads. While this paper requires less parameters in the attention layer, its construction requires large FFNs.
   - Moreover, there is no characterization on the total number of model parameters in the memorization capacity result. It's not surprising at all the Transformers can memorize. It is the upper/lower bound on the required number of parameters for memorization that makes the result interesting.
   - This paper is _not_ the first universal approximation theorem for two-layer Transformers with a self-attention of realistic size, and Section 4.2 _overclaims_ the contribution because of misinterpretation of [1]. The universal approximation result (Theorem 4.1) in [1] _is_ constructive and the construction is even simpler than this paper. The authors also claim that [1] make a particular assumption on the domain of functions. But those assumptions are only for the approximation rate result (Theorem 4.2), which is orthogonal to this paper.

- **Proof issues.** Equation 10 seems buggy. First, I think the authors should specify $\mathbf X^{(i)}\neq \mathbf X^{(j)}$ or $\mathcal V^{(i)}\neq \mathcal V^{(j)}$. Second, if it holds for some $i,j$ and $k,l$, then one can consider the same inequality for $j,i$ and $l,k$ and obtain a contradiction. So I don't see why the condition can be true.

- **Related works.** The year of [2] is marked as 2023 in the paper, but [2] was actually published in ICLR 2020. The authors should correct this to avoid misleading. In addition, I recommend the authors to also discuss [3,4,5].

- **Presentation issues.** There are many typos and minor errors in the paper, to name a few:

   - Page 3, "the Transformer block is represented as a combination of ..." $\to$ "the Transformer block is represented as a composition of ..."
   - Page 5, "this theorem indicates that one-layer Transformer does not have a memorization capacity": Please emphasize one-layer Transformer _with hardmax attention_.
   - Theorem 2, $\mathbf{W}^V$ $\to$ $\mathbf{W}^{(V)}$.
   - The appendix uses eq. $i$ and equation $i$ interchangeably. Please make it consistent.


[1] Approximation theory of transformer networks for sequence modeling.

[2] Are Transformers universal approximators of sequence-to-sequence functions?

[3] $O(n)$ Connections are Expressive Enough: Universal Approximability of Sparse Transformers.

[4] Universal Approximation Under Constraints is Possible with Transformers.

[5] Your Transformer May Not be as Powerful as You Expect.

**Questions:**

- What are the size of FFNs and the number of model parameters in the construction?

- Can you clarify the discussion on [1] in the paper?

- Can you clarify the proof issue mentioned in **weaknesses**?

[1] Approximation theory of transformer networks for sequence modeling.

---

> ### Author Response · Authors · 2023-11-18
> **Response to Reviewer mvi8**
>
> Thank you for taking the time to review our paper and sharing your insightful comments. We will address your questions below.
>
> > Q1. What are the size of FFNs and the number of model parameters in the construction?
>
> A1. The implementation [6] of the FF layer we adopted in Corollary 1 and 2 requires the number of parameters corresponding to the possible number of context ids and embedding dimension $d$.
> So, in the worst case, the number of parameters of the FF layer is $2n\binom{|\mathcal{V}|}{n} + d$.
> As for the self-attention layer, weight matrices $\boldsymbol{W}^{(O)}, \boldsymbol{W}^{(V)}, \boldsymbol{W}^{(K)}$ and $\boldsymbol{W}^{(Q)}$ all require $s + d$ parameters each. Thus, the number of parameters of the self-attention layer is $4(s+d)$.
>
> It is true that the number of parameters for the FF layer increases as the number of layers in an FF layer in Transformer models is fixed at $2$ layers.
> However, if we try to perform memorization in one-layer Transformers, it is necessary to perform contextual mapping in the self-attention layer, and therefore, we must necessarily construct a two-layer FF block that associates each context id with a label.
> If the depth of the FF layer is not limited to $2$ layers, then the implementation of FFNs with the optimal memorization capacity [7] can be used as in the previous study [8].
>
> Again, we would like to stress that our primary contribution lies in the assertion that a single-layer, single-head self-attention can already be a contextual mapping, and this conclusion cannot be derived from previous studies which treated the softmax function as an approximation of the hardmax function.
>
>
> > Q2. Can you clarify the discussion on [1] in the paper?
>
> A2. We refer to [1] as a non-constructive proof in the sense that the proof [9] of the Kolmogorov representation theorem for compact metric spaces the authors cited in [1] is non-constructive, and as a result, the inner and outer functions approximated by the FF layers in their construction are black-boxes.
> Of course, we agree that non-constructive proofs also have their advantages, such as making proofs clearer.
>
> Furthermore, we would like to point out that the construction in [1] includes an additional linear transformation layer before the 2-layer Transformer, in which input tokens are mapped to a high-dimensional space of dimension $4n^2d + 2n$.
>
> What we mean by "particular assumption on the domain of functions" is the encoded input space $\mathcal{X}^{(E)}$ that appears in Theorem 4.1 in [1].
> This space is defined by $\mathcal{X}^{(E)} = \\{  \boldsymbol{x}\ :\ x(x) \in \mathcal{I}^{(s)} \subset \mathbb{R}^d,$ where $\mathcal{I}^{(i)}, \mathcal{I}^{(j)}$ are disjoint, compact $\forall i,j \in \\{1,\dots,\tau\\} \\}$, and to our understanding, $\mathcal{I}^{(1)},\dots,\mathcal{I}^{(\tau)}$, which appear for the first time in the definition of the encoded input space, have to be given beforehand.
> If we consider universal approximation theorems for two-layer Transformers with positional encoding and continuous functions on a compact domain, then this encoded input space suffices.
>
> In contrast, we showed that $2$-layer Transformer without positional encoding is a universal approximator for continuous permutation equivariant functions on a compact domain.
> Our result can be readily extended for continuous but not necessarily permutation equivariant functions on a compact domain by using positional encoding, and at the same time is significant from the perspective of geometric deep learning.
>
> To make clearer the contributions of [1] and the novelty of our results, we have updated the discussion after Prop.1 in our paper, and changed the phrases in the abstract and "Our Contributions" part as follows.
>
> Before: universal approximation theorem for continuous functions.
>
> After: universal approximation theorem for continuous "perumtation equivariant" functions.
>
> Does our reply above answer your question? We have been given a very important point to make.
> Let us know if our answer is still incomplete.
>
>
> [1] Approximation theory of transformer networks for sequence modeling.
>
> [6] Understanding Deep Learning Requires Rethinking Generalization.
>
> [7] On the Optimal Memorization Power of ReLU Neural Networks.
>
> [8] Provable Memorization Capacity of Transformers.
>
> [9] Dimension of metric spaces and Hilbert’s problem 13.

---

> > ### Author Response · Authors · 2023-11-18
> > **Response to Reviewer mvi8 (Continued)**
> >
> > > Q3. Can you clarify the proof issue mentioned in weaknesses?
> >
> > A3. As for the bugs in equation (10), you are absolutely right. To be precise, the left-hand side of equation (10) should be its absolute value, and also the condition $\mathcal{V}^{(i)} \neq \mathcal{V}^{(j)}$ should be added.
> > Thank you for reporting the bugs.
> >
> > We also very much thank you for informing us of the presentation issues and the typo regarding the year of publication [2]. We have fixed these issues and updated our paper.
> > We have also added [3,4,5] to the related work section.
> >
> >
> > [2] Are Transformers universal approximators of sequence-to-sequence functions?
> >
> > [3]  Connections are Expressive Enough: Universal Approximability of Sparse Transformers.
> >
> > [4] Universal Approximation Under Constraints is Possible with Transformers.
> >
> > [5] Your Transformer May Not be as Powerful as You Expect.

---

> > > ### Comment · Reviewer_mvi8 · 2023-11-21
> > > **Thank you for the response**
> > >
> > > I thank the authors for the response. Some of my concerns are addressed.
> > >
> > > However, I still have concerns regarding the memorization capacity results:
> > >
> > > - The statements of the memorization capacity results are still not satisfactory. Standard memorization capacity results should characterize the required number of parameters for memorizing $N$ data samples [1,2]. The authors need to explicitly discuss this and update the statements of the Corollaries.
> > >
> > > - Besides, I have to point out the dependency on $\binom{|\mathcal V|}{n}$ is very bad (and vacuous). Can this be replaced with a linear or even sub-linear dependency on $N$? I believe the results in [3] (and other related papers) may be useful for the authors and I encourage the authors to elaborate on this.
> > >
> > > [1] Provable Memorization Capacity of Transformers.
> > >
> > > [2] Memorization Capacity of Multi-Head Attention in Transformers.
> > >
> > > [3] On the Optimal Memorization Power of ReLU Neural Networks

---

> > > > ### Author Response · Authors · 2023-11-22
> > > > **Response to Reviewer mvi8**
> > > >
> > > > Thank you very much for your prompt feedback and for further clarifying your concerns.
> > > >
> > > > > Q4. The statements of the memorization capacity results are still not satisfactory. Standard memorization capacity results should characterize the required number of parameters for memorizing $N$ data samples [1,2]. The authors need to explicitly discuss this and update the statements of the Corollaries.
> > > >
> > > > A4. We have added an evaluation of the required number of  parameters $4(s + d) + d(2nN + d)$ to the Corollaries.
> > > >
> > > > We highly appreciate your suggestive comment to help us improve our Corollaries further.
> > > >
> > > > > Q5. Besides, I have to point out the dependency on $\binom{|V|}{n}$ is very bad (and vacuous). Can this be replaced with a linear or even sub-linear dependency on $N$? I believe the results in [3] (and other related papers) may be useful for the authors and I encourage the authors to elaborate on this.
> > > >
> > > > A5. The required number of parameters for the memorization of one-layer Transformers can be evaluated as $O(N)$.
> > > >
> > > > In fact, when we focus on the order of parameters in terms of $N$, the possible number of context ids is at most $nN$. Thus, the construction [4] of the FF layer we adopt requires $2nN+d$ parameters (to be precise, $d \times (2nN + d)$ because of output dimension $d$). By adding the number of parameters in the self-attention layer, which is $4(s + d)$, the total number of parameters necessary for one-layer Transformers to memorize $N$ input-output pairs of sequences with sequence length $n$ and embedding dimension $d$ is $4(s + d) + d(2nN + d)$, which is linear dependency on $N$.
> > > >
> > > > Note that this linear dependency is optimal up to logarithmic factors for the one-hidden-layer FF block [5]. We have added a remark on the parameter efficiency after Corollary 1.
> > > >
> > > > It seems non-trivial as to whether the required parameters become sub-linear when allowing the FF layer to be deep.
> > > >
> > > > This is mainly because prior works assume that the separatedness of inputs are almost independent of the number $m$ of data. More precisely, for $(r,\delta)$-separated $m$ inputs to memorize, it is necessary that $\log (r\delta^{-1}) = \tilde{O}(1)$ [3] or $\log (r\delta^{-1}) = O(m^{2/3})$ [6].
> > > > In contrast, when attempting contextual mapping with a one-layer self-attention, context ids become $(r, \delta)$-separated with $\log (r\delta^{-1}) = O(n^4N^4)$ (equation 8), a scenario not considered in prior works [3,6].
> > > >
> > > > The question of whether there exists a deep FFN capable of achieving memorization with sub-linear parameters for datasets with such a narrow input intervals is highly interesting in itself, and could lead to a new line of research. Again, we deeply appreciate your insightful comments.
> > > >
> > > > [1] Provable Memorization Capacity of Transformers.
> > > >
> > > > [2] Memorization Capacity of Multi-Head Attention in Transformers.
> > > >
> > > > [3] On the Optimal Memorization Power of ReLU Neural Networks.
> > > >
> > > > [4] Understanding Deep Learning Requires Rethinking Generalization.
> > > >
> > > > [5] Nearly-tight VCdimension and pseudodimension bounds for piecewise linear neural networks.
> > > >
> > > > [6] Provable Memorization via Deep Neural Networks using Sub-linear Parameters.

---

> ### Author Response · Authors · 2023-11-22
> **Response to Reviewer mvi8 (Continued)**
>
> As for Q5, it is also worth mentioning that the linear dependency can be mitigated to the sub-linear $\tilde{O}(\sqrt{nN})$ when allowing for the deeper FF block, under the condition that the size $|\mathcal{V}|$ of the vocabulary set is independent of the input length $n$ and the number $N$ of input sequences.
> In such a case, the intervals $\delta$ between context ids, that is, the right-hand of equation 8 satisfies $\log \delta^{-1} = \tilde{O}(1)$, and thus the implementation of the deep FF layer from [3] can be applied.
>
> We have added this line of discussion to the Remark 2 after the Corollary 1.
>
> [3] On the Optimal Memorization Power of ReLU Neural Networks.

---

> > ### Comment · Reviewer_mvi8 · 2023-11-22
> >
> > I thank the authors for the additional discussions. My remaining concerns are addressed. I raise my rating to 6.

---

### Official Review · Reviewer_EXsH · 2023-11-01

**Soundness:** 3 good
**Presentation:** 3 good
**Contribution:** 3 good
**Rating:** 6
**Confidence:** 3

**Summary:**

The paper critically examines the expressive capacity of Transformer models, specifically addressing the discrepancy between theoretical analyses and practical implementations of Transformers. Existing analyses have often necessitated overly deep layers or numerous attention heads for data memorization, which doesn't align with the Transformers used in real-world applications. This misalignment is largely attributed to the interpretation of the softmax function as an approximation to the hardmax function.

**Strengths:**

1. The paper is well-organized and clearly written, which is easy to follow.
2. The problem studied in this paper is interesting and valuable.
3. The theoretical work of this paper is sufficient, which improves the value of the paper.

**Weaknesses:**

This paper delves into the theoretical underpinnings of one-layer Transformers. However, there are areas that could benefit from further exploration:
1. While the study provides insights into one-layer Transformers, it raises the question of whether these findings can be extended to two-layer or even deeper architectures. How scalable is the presented theory?
2. The role of the number of attention heads in determining the memorization capacity remains unclear. If it does have an impact, are there any quantitative metrics provided to elucidate its influence?

**Questions:**

Please see the Weaknesses

---

> ### Author Response · Authors · 2023-11-18
>
> Thank you for taking the time to review our paper and sharing your insightful comments. We will address your questions below.
>
> > Q1. While the study provides insights into one-layer Transformers, it raises the question of whether these findings can be extended to two-layer or even deeper architectures. How scalable is the presented theory?
>
> A1. We consider that one of the primary benefits of increasing the depth of the layers is that self-attention mechanisms may no longer need to identify the input sequence all at once.
> In other words, it becomes possible to capture the context by having self-attention mechanisms in an earlier layers recognize different parts of the input sequence, and then consolidating this information in the final layer.
>
> This computation strategy could allow the construction of models capable of calculating more separated context ids, which in turn provides a potential proof that deeper Transformers are more easily trainable.
> This paper focuses on the expressive capacity of Transformers, and while we leave this possibility on an optimization aspect of deep Transformers as a future work, we believe that delving into this research direction would be a compelling and intriguing next step.
>
>
> > Q2. The role of the number of attention heads in determining the memorization capacity remains unclear. If it does have an impact, are there any quantitative metrics provided to elucidate its influence?
>
> A2. The impact of increasing attention heads is also a highly interesting question, and we speculate that the effects may be particularly significant in real-world datasets, where inputs possess high-dimensionality.
>
> In the proof of Theorem 2, input tokens are projected into one-dimensional space using key and query matrices, and then put together to generate a context id through the softmax function. In cases where the data is scattered across multiple dimensions, however, it may be possible to focus on different dimensions with each attention head and subsequently integrate the information, like when deepening Transformers mentioned above.
> By making certain assumptions about the multidimensionality of the input distribution, it might be possible to theoretically analyze the effects of multiple attention heads. We think that this direction, too, is a very interesting future work.

---

### Official Review · Reviewer_A9Eo · 2023-11-09

**Soundness:** 4 excellent
**Presentation:** 3 good
**Contribution:** 4 excellent
**Rating:** 8
**Confidence:** 3

**Summary:**

The paper proves a new approximation results for Transformers with a single layer and attention head and dim-1 head size. The key result is that under suitable conditions on the input sequences, there exists a self-attention layer with softmax function and the settings as above such that the output tokens are bounded and separated from each other by a given distance. This allows a feedforward layer to be trained on top of it and associate to each output token a given label due to the memorization capacity of FFN layers. Hence, two theorems regarding the memorization capacity of such Transformers are proven. Further, exploiting this result, a novel universality result is proved for two-layer and single-head Transformers for permutation equivariant functions.

**Strengths:**

Very interesting paper! The discussion is surprisingly readable, although I had some issues with interpreting some keywords which are not properly introduced. The results are quite significant, since this is the first paper according to my knowledge that proves approximation results for Transformers with a single layer and attention head. It's interesting that a dimension-1 head can already produce a contextual mapping (bounded and separated output tokens), hence allowing to train an FFN on top to map each token to the required label. Detailed proofs are provided in the appendix, which I did not check completely, but from what I've checked they seem correct.

**Weaknesses:**

I guess one weakness (which does not deduct from the value of the theoretical results) is that the experiments section simply demonstrates that such a Transformer (single layer single head head dim-1 and tied weights between Q K and V) can already memorize a dataset. Maybe one thing I would be interested in is whether keeping the projection vector fixed and set to the value found using the technique in the proof of Theorem 2 using Lemma 3 can already perform well in practice?

I also had some issues with a few keywords that were not introduced, for example, in the discussion of Theorem 2 I had no idea what a sequence id was, or how it is used to construct the context id (which if I understood correctly is the output of the attention map?).

Some interesting follow-up questions were left unanswered, for example, how should the size of the FFN layer scale theoretically given the separatedness of the output tokens, or how having multiple attention heads and/or with higher rank can improve the performance?

**Questions:**

I have a couple questions:
- What is a sequence id, similarly what is a $v$-dependent sequence id?
- Is there any implication regarding the value of $\delta$ in the contextual mapping? In particular, how are the following approximation results affected if $\delta$ decreases?
- How does the memorization capacity depend on the vocabulary size?
- As is mentioned above, can we value of the rank-$1$ projection be computed as implied by the theory and kept at that value in the experiments to see how well that performs?
- Is there any difficulty with extending Prop. 1 using positional encoding to consider functions that are not permutation equivariant?

---

> ### Author Response · Authors · 2023-11-18
>
> Thank you for taking the time to review our paper and sharing your insightful comments. We will address your questions below.
>
> > Q1. What is a sequence id, similarly what is a $\boldsymbol{v}$-dependent sequence id?
>
> A1. The term "sequence id" refers to an identifier assigned to distinguish each input sequence.
> To recognize the context of a token, it is sufficient to obtain the sequence id and the token id, which is an id corresponding to the token.
> In this study, we calculate the context id using the linear combination of a sequence id and a token id. To compute the context id in a single-layer self-attention mechanism, we use each input token vector directly for the token id. The sequence id, calculated through the self-attention mechanism, is appropriately scaled and then combined with the token id through the skip connection to obtain the context id.
>
> A $\boldsymbol{v}$-dependent sequence id for some token $\boldsymbol{v} \in \mathcal{V}$, on the other hand, is a more intricate concept deeply involved in the proof.
> As mentioned earlier, the context id is generated from the linear combination of a token id and a sequence id, so in this sense, it can be regarded as a pair of the token id and the sequence id, i.e., (token id, sequence id).
> In the prior research [1], sequence ids are uniquely generated for each input sequence. However, by considering the context id as such a pair, it becomes apparent that sequence ids need not be unique. Specifically,  given some input token $\boldsymbol{v}$, it is sufficient for the identifier to distinguish each input sequence containing $\boldsymbol{v}$. We refer to this identifier as $\boldsymbol{v}$-dependent sequence id.
>
>
> > Q2. Is there any implication regarding the value of delta in the contextual mapping? In particular, how are the following approximation results affected if delta decreases?
>
> A2. The term $\delta$ does not have an effect on the number of parameters in the context of universal approximation.
> The number of weights of the feed-forward layer we adopt as an implementation [2] depends solely on the number of possible context ids and the embedding dimension $d$. It is particularly independent of the value of $\delta$.
>
>
> > Q3. How does the memorization capacity depend on the vocabulary size?
>
> A3. Unlike $\delta$, the vocabulary size has an impact on the number of parameters of the FF layer. Specifically, the maximum number of possible context ids is $n\binom{|\mathcal{V}|}{n}$. Therefore, when the implementation of the FF layer follows [2], $2n\binom{|\mathcal{V}|}{n} + d$ parameters are required for memorization.
>
>
> > Q4. As is mentioned above, can we value of the rank-$1$ projection be computed as implied by the theory and kept at that value in the experiments to see how well that performs?
>
> A4. The projection vectors obtained in Lemma 3 originates from Lemma 2, or more precisely, Lemma 13 in [3]. Unfortunately, Lemma 13 in [3] was demonstrated through a probabilistic proof, making it challenging to prepare vectors that satisfy the desired property, i.e., equation (73) before training.
>
>
> > Q5. Is there any difficulty with extending Prop. 1 using positional encoding to consider functions that are not permutation equivariant?
>
> A5. We believe it is relatively straightforward to extend Prop. 1 for continuous functions on a compact domain that are not permutation equivariant.
> In the original proof of universal approximation theorems for Transformers [4], the authors showed in Theorem 3 that Transformers with positional encoding can approximate an arbitrary continuous function on a compact domain that are not necessarily permutation equivariant with arbitrary precision. So basically the same proof technique should be applicable to our Prop. 1.
>
>
> [1] Provable Memorization Capacity of Transformers.
>
> [2] Understanding Deep Learning Requires Rethinking Generalization.
>
> [3] Provable Memorization via Deep Neural Networks Using Sub-linear Parameters.
>
> [4] Are Transformers Universal Approximators of Sequence-to-Sequence Functions?

---

### Meta-Review · Area_Chair_8Wz8 · 2023-12-25

**Metareview:**

The paper attempts to improve our theoretical understanding of transformers. In particular, it explores the expressive capacity of Transformer in the context of data memorization and universal approximation. This author show that Transformers with one self-attention layer using low-rank weight matrices, plus two FFNs, are universal appoximators of sequence-to-sequence functions. To show this it makes a connection between the softmax function and the Boltzmann operator, and argues that softmax attention can be more useful for universal approximation than hardmax which is has been used in prior papers mostly till now. Reviewers generally appreciated the paper's result, but noted weaknesses in related works, analysis w.r.t. model size, limited scope of experiments, and minor issues with the proof techniques. We thank both the authors and reviewers for engaging during the discussion phase towards improving the paper. Author response clarified some of these weaknesses. Please include all the changes requested by the reviewers including:
- it will be useful to tone down the claims in Section 4.2
- additional related works. Also for the negative result in section 3.3 with hardmax, it will be worth comparing to negative result in [1, Proposition 1].

[1] Big Bird: Transformers for Longer Sequences, NeurIPS 2020

**Justification For Why Not Higher Score:**

- Limited take away message: Authors didn't explore the consequences of their results much, e.g. number of parameters vs memorizations, etc.
- Limited empirical validation, the need for more comprehensive experiments
- Missing related work and positioning among the literature

**Justification For Why Not Lower Score:**

The paper shows expressivity for Transformers with a single layer and attention head, which is a topic of interest to the community. Also the relation between softmax function and the Boltzmann operator can be useful.

---

### Decision · Program_Chairs · 2024-01-16

Accept (poster)